# Photochemical spin-state control of binding configuration for tailoring organic color center emission in carbon nanotubes

Yu Zheng[1✉], Yulun Han[2], Braden M. Weight[1,2,3,4], Zhiwei Lin[5], Brendan J. Gifford[6], Ming Zheng [5], Dmitri Kilin [2], Svetlana Kilina[2], Stephen K. Doorn[1], Han Htoon [1✉] & Sergei Tretiak [1,6✉]

Incorporating fluorescent quantum defects in the sidewalls of semiconducting single-wall carbon nanotubes (SWCNTs) through chemical reaction is an emerging route to predictably modify nanotube electronic structures and develop advanced photonic functionality. Applications such as room-temperature single-photon emission and high-contrast bio-imaging have been advanced through aryl-functionalized SWCNTs, in which the binding configurations of the aryl group define the energies of the emitting states. However, the chemistry of binding with atomic precision at the single-bond level and tunable control over the binding configurations are yet to be achieved. Here, we explore recently reported photosynthetic protocol and find that it can control chemical binding configurations of quantum defects, which are often referred to as organic color centers, through the spin multiplicity of photo-excited intermediates. Specifically, photoexcited aromatics react with SWCNT sidewalls to undergo a singlet-state pathway in the presence of dissolved oxygen, leading to *ortho* binding configurations of the aryl group on the nanotube. In contrast, the oxygen-free photoreaction activates previously inaccessible *para* configurations through a triplet-state mechanism. These experimental results are corroborated by first principles simulations. Such spin-selective photochemistry diversifies SWCNT emission tunability by controlling the morphology of the emitting sites.

[1] Center for Integrated Nanotechnologies, Materials Physics and Applications Division, Los Alamos National Laboratory, Los Alamos, NM 87545, USA. [2] Department of Chemistry and Biochemistry, North Dakota State University, Fargo, ND 58102, USA. [3] Department of Physics, North Dakota State University, Fargo, ND 58102, USA. [4] Department of Physics and Astronomy, University of Rochester, Rochester, NY 14627, USA. [5] Materials Science and Engineering Division, National Institute of Standards and Technology, Gaithersburg, MD 20899, USA. [6] Center for Nonlinear Studies, and Theoretical Division Los Alamos National Laboratory, Los Alamos, NM 87545, USA. ✉email: yz94rice@gmail.com; htoon@lanl.gov; serg@lanl.gov

Chemical functionalization of the sidewalls of semiconducting single-wall carbon nanotubes (SWCNTs), which display structure-selective near-infrared photoluminescence[1,2], provides an emerging route to introduce fluorescent quantum defects in the nanotube that are often referred to as organic color centers (OCC). This leads to the modification of SWCNT electronic structures, achieving the adjustment of their quantum emission properties, and facilitating nanotube optical functionality including high-contrast bio-imaging, bio-sensing, photon upconversion, and single-photon emission[3–10]. The exceptional promise of SWCNT OCC quantum defects has led to a dramatic resurgence in experimental synthesis and theoretical modeling of SWCNT functionalization chemistry toward defect control[11–13]. Covalent reactions between SWCNT sidewalls and reactive agents (e.g., ozone, hypochlorite, diazonium salts, photoexcited aromatics, and guanine endoperoxide) create emissive defect-states that can trap the mobile band-edge $E_{11}$ excitons, leading to a spectrally shifted photoluminescence and an enhanced quantum yield[11–18]. SWCNTs functionalized by various aryl-based derivatives are of special interest because they have been shown to generate single photons with ultra-high purity at room temperature[7], which is essential for secure communication in modern high-speed networks and emerging quantum information technologies.

The covalent attachment of an aryl group to a single SWCNT leads to an intermediate with radical character and gives high reactivity to the three adjacent carbon sites in *ortho* positions and to the three in *para* positions (see Fig. 1)[19–21]. These secondary reactive sites can be subsequently functionalized (e.g., with a H or a OH "auxiliary group"), resulting in six topologically distinct possible aryl/H (or aryl/OH) binding configurations on the nanotube, each of which represents different $sp^3$ - hybridized defects, thus giving rise to distinct emission energies. Both H- and OH- binding are relevant to experiments typically performed in aqueous environment. Aryl functionalization is commonly achieved through diazonium chemistry[15]. However, this chemical reaction proceeds through the ground state and generates only *ortho* binding configurations[21,22]. It was shown that aryl diazonium chemistry activates solely *ortho* configurations of the near "armchair" SWCNTs such as (6, 5) and (7, 6), leading to moderate redshifts (<200 meV; denoted as $E_{11}^*$ band), whereas *ortho* configurations of the near "zigzag" nanotube species such as (9, 1) and the "zigzag" species (11, 0) cause much stronger redshifts (>200 meV; denoted as $E_{11}^{**}$ band)[22,23]. Another promising synthetic avenue proposed in a recent study[12] is a thermal reaction mechanism proceeding under the dark and under UV irradiation. Similar to diazonium chemistry, this enables formation of several *ortho* defect configurations. The use of UV light or dark conditions allows delineation between $E_{11}^*$ and $E_{11}^{**}$ emissions, in particular, leading to highly pure $E_{11}^{**}$ spectral features thus demonstrating a remarkable synthetic control of the resultant two *ortho*- binding configurations. While feasibility of *para* defects was suggested by theoretical simulations[21,23], practical chemical routes have so far been limited to *ortho* configurations, hindering SWCNT photoluminescence tunability. As illustrated in Fig. 1, photochemical reactions may offer new pathways, since energy barriers experienced by reactants in their ground state potential energy surface (PES) can be reduced or completely waived in an excited state PES with different spin multiplicity, which enables additional binding configurations[24–29].

Here, we demonstrate that recently developed synthetic protocol is capable of controlling binding configurations by accessing the previously inaccessible *para* geometry of quantum defects in SWCNTs through a precise photochemical functionalization. These new photochemical pathways can be enabled and controlled via oxygen excess. This synthetic route - relying on photoexcited chemistry - was first reported by a few of us (see ref. [18]). This work hypothesized that different spectral features arise from the formation of different binding configurations but was unable to definitively assign which configurations had been formed. In this contribution we identify that chemical reactions between (6, 5) SWCNT sidewalls and photoexcited aromatics in the presence of dissolved oxygen generate additional emission features redshifted by ca. 160 meV (*i.e.*, $E_{11}^*$ transition), whereas the oxygen-free photoreaction gives two distinct pronounced emission bands red-shifted by ca. 140 and 260 meV (*i.e.*, $E_{11}^*$ and $E_{11}^{**}$ transitions, respectively), originating from *ortho* and *para* binding configurations, respectively. The previously unachievable $E_{11}^*$ emission for (11, 0) SWCNTs that arises from the *para* configuration is also obtained through the oxygen-free photochemistry. The observed trend is rationalized based on the fact that oxygen molecules are efficient triplet quenchers. Therefore, photoexcited aromatics react with SWCNT sidewalls via a singlet pathway in air-saturated samples, activating *ortho* binding configurations of aryl/H in the nanotube. In the absence of oxygen, photoexcited aromatics are covalently bound to nanotube surfaces undergoing either singlet or triplet pathways, thus activating *ortho* or *para* geometries, respectively. To understand the atomistic mechanism of these photo-activated reactions and their dependence on spin multiplicity, we have modeled the reaction with and without triplet quenching using spin-constrained density functional theory (DFT)[30]. Computed reaction pathways demonstrate selectivity and dependence of outcomes of the photo-reaction on the deactivated or activated triplet pathway, corresponding to presence or absence of oxygen, respectively. In particular, our simulations reveal that binding to *para* configurations through the triplet state exhibits a lower energy barrier than that via the singlet state at the transition state geometry. In contrast, the situation is reversed for *ortho* binding configurations: Here the barriers for triplet state reactions are higher than their singlet state counterparts. These computational findings can be rationalized with the Pauli exclusion principle suggesting preferable spatial separation of reactive electrons in the triplet state leading

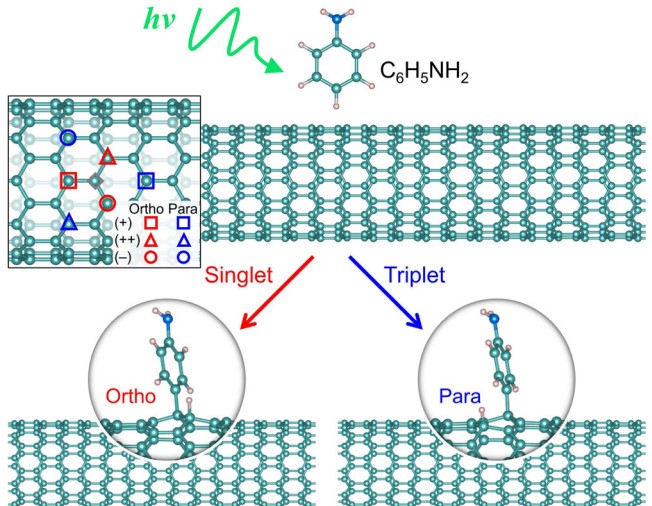

**Fig. 1 Schematic diagram illustrating the spin-state controlled formation of $sp^3$-defect in SWCNTs.** The insert shows six possible ways of creating a single $sp^3$-defect with the gray diamond symbol representing the position of aryl group attachment on the nanotube surface. The red and blue symbols identify the secondary reactive sites for H atom, leading to three *ortho* (red) and three *para* (blue) defect geometries, each of which is named as (+), (++), and (−) depending on the direction of the bond involved in the formation of the $sp^3$-defect with respect to the nanotube surface. In shown example of (11,0) SWCNT, only degenerate *ortho* (−) and (++) configurations are shown to be enabled by diazonium chemistry[22].

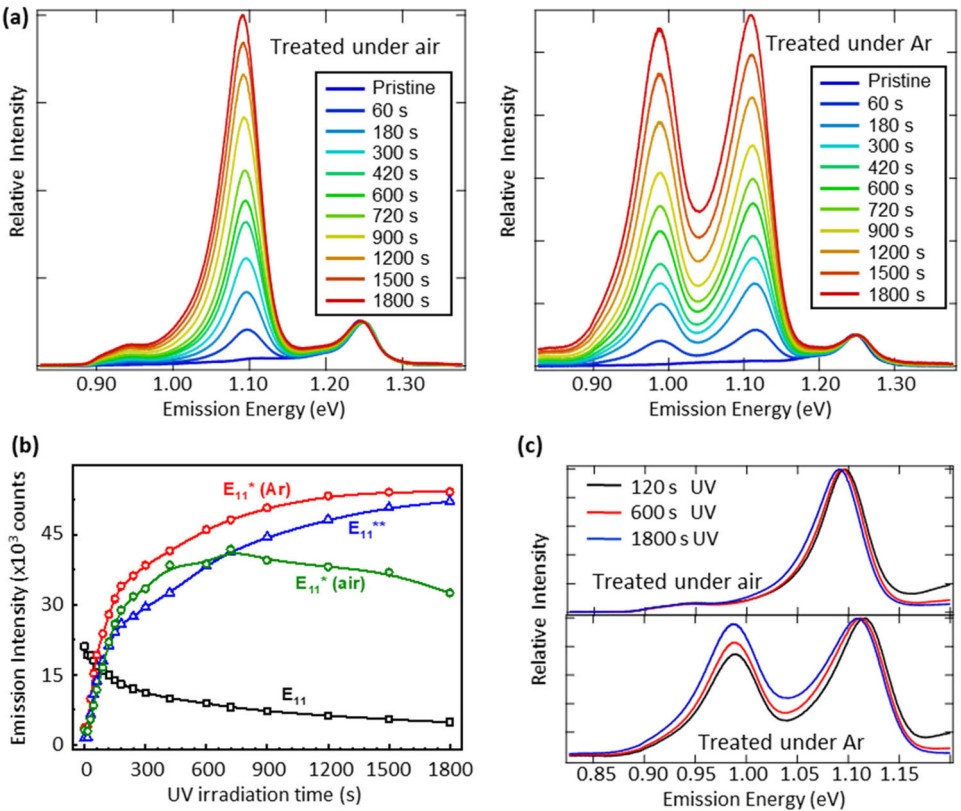

**Fig. 2 Kinetics of defect-state photoluminescence. a** PL spectra (excited at 570 nm) of (6,5) SWCNT samples containing 1 mM *p*-iodoaniline and irradiated with UV light for different times under ambient (left panel) and argon (right panel) conditions. **b** Emission intensity *vs* UV irradiation time for (6,5) samples containing 1 mM *p*-iodoaniline. The black curve shows the pristine emission ($E_{11}$). The green curve represents $E_{11}^*$ defect-state emission obtained under ambient conditions. The red and blue curves show defect-state emission $E_{11}^*$ and $E_{11}^{**}$, respectively (obtained under argon conditions). **c** Prolonged UV irradiation causes slight red-shifts of defect-state emission.

to *para* configurations. The attained understanding introduces an important and distinctive perspective to SWCNT chemistry utilizing spin states in photoreactions to control binding selectivity and achieve previously inaccessible *para* configurations. Such spin-selective photochemistry develops a new landscape to tailor the electronic structures of π-electron conjugated materials and predictably modify their optical properties at an atomistic level.

## Results

**Synthesis and spectroscopic features.** We introduce fluorescent quantum defects in SWCNT surfaces through a photoreaction between the nanotube sidewalls and aromatic reagents that are excited by UV irradiation. Photoexcited aromatics (e.g., *p*-iodoaniline, aniline, and nitrobenzene) allow aryl groups to covalently bind to the nanotube sidewalls, causing enhanced photoluminescence (PL) quantum yield and red-shifted emission from defect-states (see Fig. 2 and Supplementary Fig. 1)[18]. Energies of emitting states at defect sites strongly depend on the availability of dissolved oxygen during the photoreaction. The defect-state emission feature red-shifted by 160 meV (i.e. $E_{11}^*$ transition) appears during the photoreaction in air-saturated (6, 5) SWCNT samples, whereas the oxygen-free reaction in an argon (Ar) atmosphere displays growth of two emission bands red-shifted by 137 and 260 meV (i.e. $E_{11}^*$ and $E_{11}^{**}$ transitions, respectively), as shown in Fig. 2a. These results are consistent with spectral changes for an ensemble of functionalized CoMo-CAT SWCNTs prepared using the same synthetic protocol[18]. Control experiments reported previously (Figs. S10-S12 of Supporting Information in ref. [18]) confirm that the covalent SWCNT functionalization indeed requires addition of aromatic

compounds followed by UV photoexcitation. This delineates present SWCNT functionalization from oxygen doping achieved by exposure of nanotubes to reactive oxygen species generated by exposure of aqueous samples to high-energy UV photons[31].

The photoreaction progress was controlled by the duration of UV light exposure under a fixed power (ca. 6 mW /cm²) and was monitored by tracking PL spectral changes. The intensities of emission from band-edge $E_{11}$ and defect-states $E_{11}^*$ (air), $E_{11}^*$ (Ar), and $E_{11}^{**}$ (Ar) are plotted as functions of UV irradiation time in Fig. 2b. The pristine emission decreases monotonically as the reaction proceeds, accompanied by a concomitant increase in defect-state emission. Prolonged irradiation in air-saturated samples causes a decrease in $E_{11}^*$ defect-state emission while emission features from defect-states that are formed through the oxygen-free reaction keep growing during the 30 min irradiation. The simultaneous growth of $E_{11}^*$ and $E_{11}^{**}$ emission peaks during the oxygen-free photoreaction implies that the aryl functionalization of SWCNTs proceeds via two parallel reaction channels, producing different defect geometries in the nanotube and thus leading to distinct emitting defect states. Prolonged irradiation in the presence of oxygen can only cause slightly red-shifted $E_{11}^*$ emission without any appearance of $E_{11}^{**}$ band (see Fig. 2c). This contrasts with the observations in ground-state diazonium chemistry where $E_{11}^{**}$ emission arises at high concentration of diazonium species leading to functionalization[32]. Therefore, the oxygen-involved photoreaction selectively introduces specific binding configurations of aryl defects whose energies correspond to $E_{11}^*$ emitting states, whereas the defect geometries with energies corresponding to $E_{11}^{**}$ emitting states can only be formed through the photochemistry in the absence of

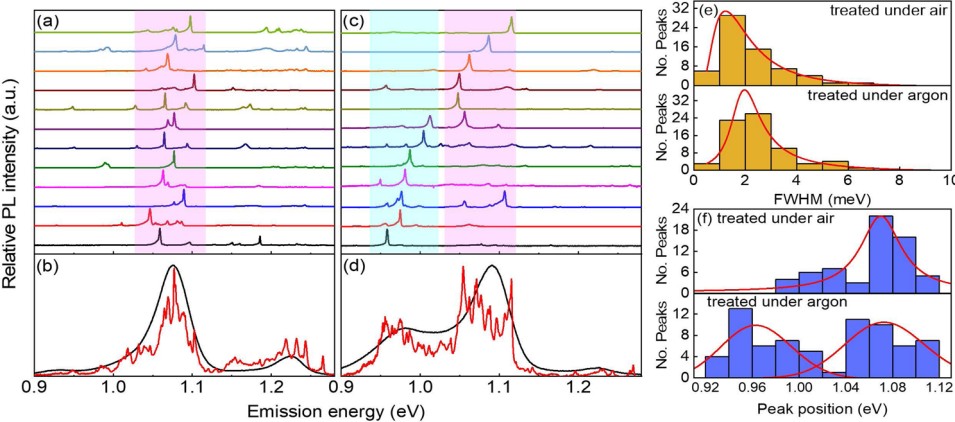

**Fig. 3 Low-temperature single photoluminescence spectra. a** Low-temperature PL spectra (at ~ 4 K) for 12 individual (6,5) SWCNTs functionalized through the oxygen-involved photoreaction. **b** The black curve shows emission from the solution-phase (6,5) sample treated under air conditions. The red curve represents the average low-T PL spectrum of 50 individual SWCNTs in **a**. **c** Low-T PL spectra for (6,5) SWCNTs functionalized through the oxygen-free photoreaction. **d** The black curve shows emission from the solution-phase sample treated under argon conditions. The red curve is the average low-T PL spectrum of 50 individual SWCNTs in **c**. **e** Histograms showing distributions of PL peak linewidths (FWHM) for (6,5) SWCNTs treated under ambient (up frame) and argon (bottom frame) conditions. **f** Histograms showing distributions of PL peak positions for (6,5) SWCNTs treated under ambient (up frame) and argon (bottom frame) conditions.

oxygen. We expect that all photochemical reactions ultimately lead to the true ground state with singlet multiplicity for all functionalized SWCNT species, which later is probed spectroscopically.

**Low-temperature spectroscopy.** Low-temperature single PL spectra were obtained for individual SWCNTs photochemically functionalized in the presence and absence of oxygen. Single tube PL spectra at ~ 4 K for 12 out of ~ 50 individual (6, 5) SWCNTs functionalized under air (see Fig. 3a) and under argon (see Fig. 3c) display strong, sharp, and asymmetrical peaks in the range of ca. 1100–1200 nm (1.13–1.03 eV) and ca. 1100–1300 nm (1.13–0.95 eV), respectively. Aryl functionalization via the oxygen-involved photoreaction leads to inhomogeneous optically active defect states with energies spanning from 0.98 to 1.12 eV, which originate from different dielectric environments for each individual nanotubes due to different interactions with the substrate (see Fig. 3a, f). The accumulated average of those sharp single PL spectra agrees with the ensemble spectrum for the solution-phase sample at room temperature, showing a broad $E_{11}^*$ defect-state emission feature (see Fig. 3b).

The low-temperature single-nanotube PL spectra for aryl-functionalized SWCNTs obtained via oxygen-free photoreaction display three spectral patterns including emission only in the 1100–1200 nm (1.13–1.03 eV) spectral range (type I), emission only in the range of 1200–1300 nm (1.03–0.95 eV) (type II), and multiple pronounced emission peaks in both ranges (1.13–0.95 eV) (type III) (see Fig. 3c, f and Supplementary Fig. 2). The emitting states with energies of 1.03–1.13 eV are assigned to $E_{11}^*$ defect-states while the emitting states in the 0.95–1.03 eV energy range arise from $E_{11}^{**}$ defect states. The accumulated average of single nanotube emission agrees with the ensemble photoluminescence spectrum for the solution-phase sample, displaying two distinct defect-state emission bands, $E_{11}^*$ and $E_{11}^{**}$. Given the coexistence of well-pronounced type I and type II optical features in an aryl-functionalized sample, we believe that $E_{11}^*$ and $E_{11}^{**}$ emitting states are formed independently, indicating that distinct binding configurations of aryl/H are established in parallel through photochemistry in the absence of oxygen. The existence of type III features signifies that $E_{11}^*$ and $E_{11}^{**}$ defect-states can also be formed simultaneously on the same single nanotube in oxygen-free conditions. We have

observed that the $E_{11}^*$ defect-state emission obtained under air is red-shifted by ca. 20 meV from that formed in the absence of oxygen, whereas linewidths of single PL peaks from $E_{11}^*$ (air), $E_{11}^*$ (Ar), and $E_{11}^{**}$ (Ar) defect-states are similar with an average value of ca. 2.5 meV (see Fig. 3e, f and Supplementary Fig. 2b and Supplementary Table 1).

**Analyses of multiple nanotube chiralities.** To further explore the SWCNT structure-dependent emitting defect-states that are formed via the photochemistry, we prepared samples of highly enriched single nanotube chiralities with diameter ranging from 0.692 to 0.936 nm and with chiral angle ranging from 0 (zigzag) to 27.5° (near armchair). By considering a diverse family of SWCNTs, we aim to associate spectral features appearing due to photochemical reactions with that formed through conventional diazonium chemistry (which underpin *ortho* configurations), as well as to identify additional unique emissive features stemming from *para* defects (Fig. 4a). The near-armchair nanotube species such as (6,4), (6,5), (7,5) and (7,6) display only $E_{11}^*$ defect-state emission after the oxygen-involved photoreactions (see Supplementary Figs. 3–6). This is consistent with the observations of aryl-functionalization through conventional diazonium chemistry that leads to *ortho* binding configurations only and activates predominantly $E_{11}^*$ emission in the nanotube[22]. In addition, the oxygen-free photoreactions activate the $E_{11}^{**}$ emitting states for these near-armchair nanotube structures, which we assign to previously inaccessible *para* binding configurations (see Fig. 4 and Supplementary Figs. 3–6). Former aryl-functionalization methods only allow $E_{11}^{**}$ defect-state emission for the near zigzag and zigzag nanotube species such as (9, 1) and (11, 0), originating from *ortho* binding configurations[22]. Here the $E_{11}^*$ emission is suppressed. The previously inaccessible $E_{11}^*$ emission for those near zigzag species is now obtained via photochemistry in the absence of oxygen (see Fig. 4a and Supplementary Fig. 7). Therefore, the oxygen-free photoreaction allows the formation of commonly unachievable emitting states in SWCNTs, whereas the oxygen-involved photochemistry generates emitting states similar to previous chemical methods.

Given the pronounced $E_{11}^*$ and $E_{11}^{**}$ emission attained through the precise photochemistry process virtually for every chirality, we further analyze the energy shifts of these emitting defect-states with respect to their band-edge excitons ($E_{11}$) that

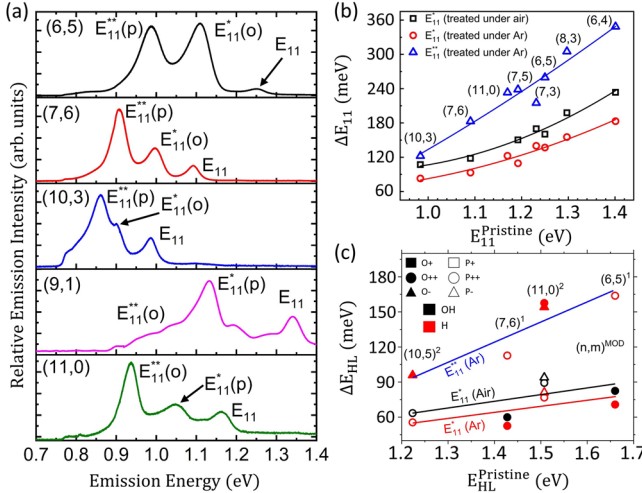

**Fig. 4 Defect-state photoluminescence spectra for different chiralities.**
**a** PL spectra for single-chirality enriched samples functionalized through the oxygen-free photoreaction. The black curve shows emission from (6,5) (excited at 570 nm). The red curve shows emission from (7,6) (excited at 662 nm). The blue curve is emission from (10,3) (excited at 640 nm). The pink curve is emission from (9,1) (excited at 700 nm). The green curve is emission from (11,0) (excited at 760 nm). $E_{11}^*(o)$ and $E_{11}^{**}(o)$ represents defect-state emission arising from *ortho* binding configurations, while $E_{11}^*(p)$ and $E_{11}^{**}(p)$ are defect-state emission arising from *para* binding configurations. **b** Energy shifts of defect-state emission from band-edge exciton *vs* pristine emission energy. Different $E_{11}$ energies correspond to different SWCNT chiralities that are labeled in the figure. Black squares represent energy shifts of $E_{11}^*$ defect-state emission obtained through the oxygen-involved photoreaction. Red circles and blue triangles represent energy shifts of $E_{11}^*$ and $E_{11}^{**}$ emission, respectively (obtained under argon conditions). **c** Ground state HOMO-LUMO (HL) energy shifts $\Delta E_{HL} = E_{HL}^{Pristine} - E_{HL}^{Defect}$ as a function of HL energy in pristine nanotube, calculated using DFT. The color of the symbols represents the chemical composition of suggested auxiliary group: Aryl-$NH_2$/H (red) in argon and aryl-$NH_2$/OH (black) in Air. The symbol shape and fill-type represent the defect configurations: *ortho*(++) (filled circles), *ortho*(−) (filled triangles), *para* (++) (open circles), and *para*(−) (open triangles). The fitted curves demonstrate the same trends as seen in **b**, where the deviation between $E_{11}^*$ energies in air and argon is attributed to the auxiliary attachment of either H or OH groups.

are plotted as functions of $E_{11}$ energy, as shown in Fig. 4b. The energy shifts of both $E_{11}^*$ and $E_{11}^{**}$ defect-states show a strong positive dependence on $E_{11}$ energy, with $\Delta E_{11}^*$ increased by 120% and $\Delta E_{11}^{**}$ increased by 185% for $E_{11}$ energy increasing from 0.98 to 1.4 eV by going from (10, 3) to (6,4) species. Stronger dependence of the energy shifts of $E_{11}^{**}$ states can be rationalized by the deeper trapping potential of $E_{11}^{**}$ defect-states that entails stronger exciton wavefunction confinement than $E_{11}^*$ defect-states, so the shifts of $E_{11}^{**}$ emitting states are more sensitive to SWCNT chirality. The $\Delta E_{11}^*$ obtained in air and argon shows a similar trend on nanotube band gap except for a systematically larger value of $\Delta E_{11}^*$ (air) than $\Delta E_{11}^*$ (Ar). The difference of $\Delta E_{11}^*$ (air) and $\Delta E_{11}^*$ (Ar) is between ca. 20 to 50 meV and shows some minor dependence on nanotube structure (see details in Supplementary Figs. 8 and 9).

Thus, the defect-state emission features - from aryl functionalization via oxygen-involved photochemistry - are similar to those obtained through diazonium chemistry. The latter has been demonstrated to favor the *ortho* binding configurations of aryl/H or aryl/OH adducts giving rise to the $E_{11}^*$ and $E_{11}^{**}$ emitting features for near-armchair and near-zigzag SWCNTs, respectively[22].

As such, it is suggestive that the emitting defect-states formed from the photoreaction in the presence of oxygen also give arise to the formation of *ortho* binding configurations. While the oxygen-free photoreactions retain the presence of these emitting states, there are new pronounced complementary $E_{11}^{**}$ or $E_{11}^*$ emission features for near-armchair and near-zigzag SWCNTs, respectively, which were not observed previously. The difference in the chirality-dependent trends between $E_{11}^*$ and $E_{11}^{**}$ produced in oxygen-free photoreactions points to their different origins that we associate with the structurally different *ortho* and *para* defect types (see Fig. 4b) that form through the singlet and triplet state photochemistry, respectively. Spectral characteristics for $E_{11}$, $E_{11}^*$, and $E_{11}^{**}$ emission of eight different nanotube chiralities are summarized in Supplementary Table 2.

**Modeling of spectroscopic features and reaction pathways.** To substantiate our assignments of experimentally observed emission features to the specific binding configurations, we further performed DFT simulations of the HOMO-LUMO gaps (see Methods) of several experimentally relevant nanotube chiralities. Overall, the DFT results (Fig. 4c) demonstrate similar trends as found in Fig. 4b, further allowing assignment of emission features to the specific binding geometries. For example, for zigzag (11,0) SWCNTs, $E_{11}^{**}$ can be prescribed to the largely red-shifted and degenerate *ortho*(++)/*ortho*(−) pair, and the newly-accessible $E_{11}^*$ state can be attributed to the moderately redshifted degenerate *para*(++)/*para*(−) pair, respectively. For near-armchair (6,5) system, $E_{11}^{**}$ is attributed to the *para*(++) configuration, leaving $E_{11}^*$ as the corresponding *ortho*(++). These peak assignments are performed by comparing calculated gaps to the relative energetics of the experimental spectra between $E_{11}$, $E_{11}^*$, and $E_{11}^{**}$ as well as to results of our previous theoretical studies[22]. In particular, we assume that *ortho*(+) is experimentally irrelevant due to its small π-orbital mismatch, leaving this configuration unreactive[32]. Moreover, our computations further attribute a minor red-shift between experimental $E_{11}^*$ emission features in argon and oxygen to the slightly different functional groups in these environments, which are Aryl-$NH_2$/H and Aryl-$NH_2$/OH, respectively (where H and OH are the defect's auxiliary groups). The effect of chemical composition of defects has been studied in previous reports and the splitting between H and OH auxiliary groups is expected to increase with inclusion of excitonic effects and with increasing basis set size[11,33]. Our computational results for all possible defect configurations are shown in Supplementary Fig. 8a.

To rationalize why triplet and singlet state photochemistry predominantly produce *para* and *ortho* binding configurations, we further computationally examine the respective reaction pathways in a (11, 0) zigzag SWCNT using DFT-based time-dependent excited state molecular dynamics approach[25,26] and climbing-image nudged elastic band (CI-NEB)[34,35] (see Methods). The calculated optical transitions of the (11, 0) SWCNT functionalized by Aryl-$NH_2$/H mainly contribute to two pronounced peaks, shifted by ca. 100 and 150 meV that originate from a set of degenerate two *para* and two *ortho* defects, respectively (Fig. 5a). Notably, the high numerical cost of reaction path sampling necessitates using a small computational cell and a semi-local DFT model (see Methods). This provides only qualitative values for optical transitions, which remain consistent with more elaborate simulations[21] and with Fig. 4c. Keeping this in mind, we assign $E_{11}^*$, and $E_{11}^{**}$ emissive peaks to *para*(++)/(−) and *ortho*(++)/(−) configurations, respectively (see Fig. 1 and Fig. 5a). The CI-NEB method provides the minimum energy path starting with an aniline molecule near but unbound to the pristine tube, and ending with one of the four

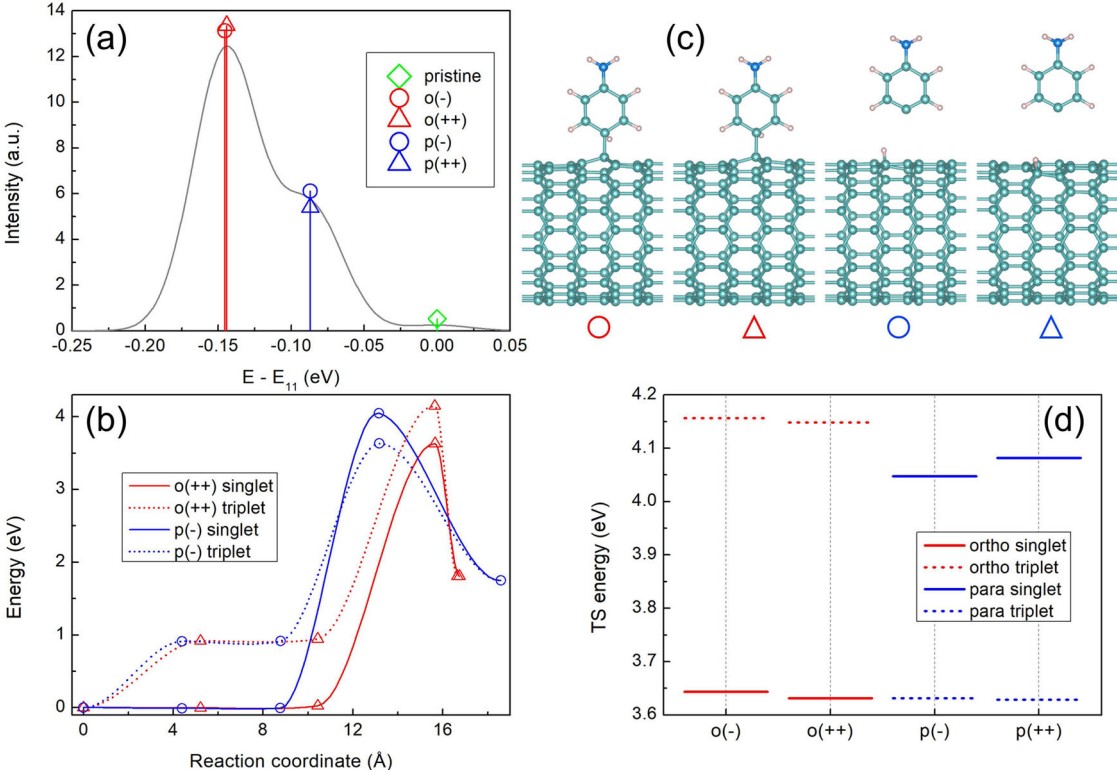

**Fig. 5 Theoretical modeling of the photoreaction pathways. a** Calculated energy shifts of HOMO-LUMO transitions for *ortho*(++)/(−) (red), *para*(++)/(−) (blue) functionalized (11,0) SWCNT from the respective pristine (green) value. These *para* and *ortho* configurations are associated with $E_{11}$* and $E_{11}$** emission features, respectively. **b** Calculated minimum energy paths leading to *para* (−) and *ortho* (++) binding configurations through singlet and triplet state photochemistry. Maximal energy for each path corresponds to the transition state. **c** Geometries of transition states leading to four binding configurations. **d** Comparison of the energy barriers at transition states calculated for singlet and triplet reaction pathways leading to four binding configurations considered.

different Aryl-$NH_2$/H binding configurations on the tube, see Supplementary Movies 1 and 2. Along the path, the method generates three intermediates, in which the one with the maximum energy represents a saddle point (*i.e.*, transition state geometry) on the potential energy surface. Figure 5b, c summarize calculated reaction paths and respective transition state (TS) structures, respectively, that the system passes during CI-NEB calculations. A detailed analyses of reaction paths and TS structures is given in Supplementary Figs. 10-16 along with a Supplementary Discussion. We note that the energy barrier heights for *ortho* (++) defects are different when the reaction follows singlet and triplet state pathways (Fig. 5b and Supplementary Fig. 12). Namely, the triplet TS is about 500 meV higher than the singlet TS, suggesting that *ortho* binding configurations are predominately accessed via singlet-state photochemistry. The situation is completely reversed for the *para* (−) defect, suggesting preferential triplet-state reactive pathways (Fig. 5b and Supplementary Fig. 13). These calculated differences in the TS energy barriers are summarized in Fig. 5d for all four defect configurations considered. Interestingly, the TS geometries leading to *para*(++)/(−) defects, which form the new $E_{11}$* emission peak, have a distinct dynamical feature: the first step in the reaction is that the H atom is transferred to the tube, while the aryl group remains far away (~4 Å) and unbonded (see Fig. 5c and Supplementary Fig. 13). The opposite situation is found for *ortho*(++)/(−) defects that contribute to the previously observed $E_{11}$** emission peak. Here, both the H and aryl group are close to the tube at their TS, but the aryl bond establishes itself first. Observed hydrogen detachment is a direct outcome of our computational modeling. These results are fully consistent

with the mechanism of many well-documented small molecule photoaddition reactions[36]. Altogether, our calculations show that previously inaccessible $E_{11}$* emitting states - corresponding to *para* binding configurations of Aryl-$NH_2$/H in the (11,0) nanotube - are achievable via triplet state photochemistry. In contrast, *ortho* defects are likely formed via the singlet state reactions.

Thus, our experimental observations and modeling results demonstrate that the photochemistry mechanism is related to the spin states of reactants, and the singlet and triplet state pathways predominantly introduce aryl defects with *ortho* and *para* binding configurations, respectively. Irradiating the aromatics and SWCNT mixtures leads to the formation of excited singlet states for aromatics, and such aromatics react with nanotube sidewalls, forming aryl defects with the *ortho* binding configuration. In addition, the excited triplet states of aromatics can be formed from the transition of the excited singlet states via the intersystem crossing process[37,38]. Since oxygen is well known as an efficient triplet quencher, its presence rapidly quenches these triplet excitations and precludes the triplet pathways of reactions. In the absence of oxygen, however, the triplet reaction pathway becomes available, which introduces aryl defects with the *para* configuration in SWCNTs and allows formation of previously unachievable emitting states. These observations can be rationalized with simple physical arguments: Singlet-state photochemistry involves a pair of electrons of opposite spins that prefer to be in a close proximity on the same orbital to maintain chemical bonding, leading to *ortho* defects anchored to the same C-C bond. In contrast, triplet-state photochemistry involves electrons of co-linear spins that tend to occupy different orbitals and be spatially separated, which results in a formation of bonds on the opposite

ends of the ring, being *para* configurations. This qualitatively explains different energetic barriers for singlet and triplet reactions pathways resulting in *ortho* and *para* binding configurations, respectively, conforming to the Pauli exclusion principle.

## Discussion

Fluorescent quantum defects (SWCNT organic color centers) are a new class of synthetic quantum emitters that can be synthetically created in SWCNTs by covalently attaching organic functional groups. A wide variety of possible bonding patterns on the carbon lattice lends to unique opportunities to precisely tailor these defects to the desirable electronic properties. This requires establishing structure-property relationships enabling synthetic routes toward atomic precision. We report here that the photochemical activation of aryl-to-SWCNT binding provides a controllable route to introduce fluorescent quantum defects in SWCNTs with specific binding configurations. The presence and absence of dissolved oxygen in aqueous SWCNT suspensions controls the binding configurations of aryl defects and thus allows adjusting SWCNT photoluminescence tunability by activating distinct emitting states from aryl defects with different binding configurations in single nanotubes.

It is important to highlight that the binding configuration is a key factor determining the emission wavelength of the fluorescent covalent defect. Our work reports several ways to control the properties of resulting emitters toward on establishing tunability of SWCNT covalent fluorescent defects. First, we have identified and demonstrated the correlation between spin multiplicity of transients and product structure. Specifically, the photochemistry undergoing the singlet pathway enables the *ortho*-only binding configuration of aryl defects in SWCNTs, whereas the previously inaccessible *para* configurations are activated through the triplet mechanism. This dependence on spin multiplicity of transition state is justified by state-of-the-art quantum chemical simulations. Our modeling shows that a constraint to spin multiplicity of reactants and transition states selects the energetically preferable photoreaction pathway and controls configurational diversity in binding chemistry, supporting and rationalizing experimental observations. Second, we have identified a strong dependence of the most red-shifted emitting states on SWCNT chirality. This observation is rationalized by the fact that the defect-states with largest red-shifts in energy show stronger sensitivity to nanotube structure than the defect-states with moderate red-shifts, arising from larger exciton confinement at deeper trapping potentials. Finally, we point out a strong chiral angle dependent photochemical reaction outcome (see Supplementary Fig. 17 and Fig. 3e in ref. [18]), which suggests presence of an additional mechanism. These trends will be explored in depth in the future studies.

The reported findings have a twofold significance. First, they impact practical applications of quantum light sources for quantum information technologies as the controlled photoreaction activates the most red-shifted defect-states and enables a variety of nanotube chiralities emitting at telecom wavelengths. Second, these findings have potential for fundamental impact beyond specific application. This concept of spin-dependent photoreaction control could be expanded to chemical functionalization of any nanomaterials, whose product structures can be precisely controlled via photochemical reactions utilizing spin states.

Altogether, this work reveals the fundamental mechanism of nanotube photochemical reactions, advances our understanding of binding chemistry, and brings SWCNT chemistry using spin multiplicity to a level that has not been achieved previously.

## Methods

**Preparation of sorted SWCNT solutions**. All photoluminescence spectra measured in this work were obtained from single-chirality highly enriched SWCNT samples. They were prepared by either the aqueous surfactant-based two-phase (ATP) sorting method[39] or the DNA-based ATP separation method[40,41]. All sorted SWCNT samples were stored in 1% (w/v) sodium deoxycholate (SDC) environment. Before the photochemistry-induced aryl-functionalization, the SDC-suspended single-chirality enriched samples were exchanged into 1% (w/v) sodium dodecyl sulfate (SDS) solutions by centrifugation filtration with a 100 kDa filter at 730 g in a bench-top centrifuge (Clay Adams Dynac Centrifuge, Model 0101). All materials were purchased from Sigma-Aldrich.

**Functionalization of suspended SWCNT samples**. The stock solutions of organic aromatic compounds (*p*-iodoaniline, aniline, and nitrobenzene) were dispersed in acetone. The concentration of those stock solutions was fixed to 50 mM. Typically, 10 µL of the aromatic stock solution was added to a 500 µL SWCNT sample. The mixtures were either treated in air conditions or purged with argon to remove dissolved oxygen in the solution. UV LED (Boston Electronics, product code VPC1A1) with emission peaking at 300 nm (280–320 nm range) was used to irradiate the aromatics/SWCNT mixtures at the power density of ca. 6 mW/cm² at a LED drive current of 0.15 A. The aromatics/SWCNT mixtures was hold in a 1 cm×1 cm UV grade quartz cuvette that was placed next to the LED ensuring that entire mixture was illuminated uniformly. The exposure times varied from 30 s to hours as indicated in the figures.

**Preparation of SWCNT substrates**. Aqueous SWCNT solutions were drop-cast on substrates for single-particle photoluminescence spectra. Glass coverslips were coated with a 300 nm gold layer using electron-beam evaporation. Polystyrene was dispersed in toluene with a concentration of 1% (w/v). Such prepared polymer solution was then spin-coated onto the gold layer. The polymer coated gold layer was used to enhance the PL collection efficiency without modifying the emission properties of SWCNT through plasmonic effects. Then a functionalized SWCNT sample was drop-cast and evaporated on the substrate. We used SWCNT solutions with relatively low concentrations to ensure isolated individual nanotubes on substrates. Before optical characterization, we washed the deposited samples with water or isopropyl alcohol serval times.

**Optical characterization**. Photoluminescence spectra of solution-phase SWCNT samples were measured by a Horiba Nanolog spectrofluorometer with an 850 nm long-pass filter in the collection path. $E_{22}$ resonant excitation wavelengths were chosen to measure different single-chirality SWCNT samples to ensure strong $E_{11}$ emission signals. Single-nanotube measurements were performed at ~ 4 K using a custom-built micro-spectrofluorometer system. Photoluminescence spectra and images were excited by a continuous-wave laser with the wavelength at 840 nm and captured by a one-dimensional InGaAs linear array detector and a two-dimensional InGaAs array camera, respectively.

**Computational simulations of band gaps**. The finite-size SWCNT calculations shown in Fig. 4 and Supplementary Figs. 8, 9, and 10 were computed using the Gaussian 16 package[42] following previously developed modeling approaches[21]. These simulations provided the ground state optimal geometry and electronic configuration with singlet multiplicity being the lowest electronic state of all functionalized nanotubes directly relevant to the experimental spectroscopy. The HOMO-LUMO calculations were performed using the B3LYP functional combined with STO-3G basis set, which has been shown to reproduce the electronic structure of sp³-hybridized SWCNTs well[43,44]. For each finite-size SWCNT, the ends were capped following a previously used scheme to reproduce the semi-infinite electronic structure[44]. The relative geometry information is provided in Supplementary Table 3. The average splitting between argon and air $E_{11}^*$ energies is 10 meV, which is smaller than seen in experiment (Fig. 4b). This is expected as the DFT calculations were performed with minimal basis set and without excitonic effects, both of which contribute to the reduction in splitting between these features, as we have shown in our previous reports[33] and in Supplementary Fig. 8b, c.

**Photochemistry modeling**. Our proof-of-concept calculations of photoreactions were performed by time-dependent excited state molecular dynamics approach[24–26,45]. Due to numerical expense, such simulations were only currently tractable for (11,0) zigzag nanotube from the family considered above, which has a small unit cell. To explore the reaction pathways, we applied the climbing-image nudged elastic band (CI-NEB) method[34,35] as implemented in Vienna ab initio simulation package (VASP)[46,47]. DFT calculations were carried out using VASP with the generalized gradient approximation (GGA) Perdew−Burke−Ernzerhof (PBE)[48] functional in a plane-wave basis set along with projector augmented-wave (PAW) pseudopotentials[49]. We used a plane-wave cutoff of 400 eV and optimized the atomistic models until the total energy was converged to an accuracy of 1×10⁻⁶ eV with a force tolerance of 0.01 eV/Å. All calculations were performed at the Γ point. There were five types of carbon nanotube containing models: one reactant with the aniline molecule placed ~ 5 Å away from the tube and four products (i.e., *para*(++)/(−) and *ortho*(++)/(−) Aryl-NH₂/H binding configurations). We

generated (11,0) CNTs with 6 unit cells (264 atoms) for CI-NEB calculations and 30 unit cells (1320 atoms) for emission calculations. We extended the length along the tubular axis to infinite under periodic boundary conditions. Vacuum spacing layers of 9 Å were added in the x and y directions to minimize any spurious interaction. The overall size of the rectangular simulation cell was defined by the lattice parameters x = 24.45 Å, y = 17.63 Å, and z = 25.56 Å or z = 127.80 Å. The CI-NEB calculations were carried out for each of the four distinct Aryl-$NH_2$/H binding positions with a force tolerance of 0.03 eV/Å. Such calculations generated three intermediates along the potential energy surface using the optimized reactant and final product as inputs under different spin multiplicity with spin-polarized DFT.

For approximate evaluation of optical spectra, the oscillator strengths for HOMO-LUMO transitions were obtained by adopting independent orbital approximations[26,50]. The HOMO-LUMO gap mainly follows the trends and characteristics of the low-lying excitons, $\Delta E_{11}*$[33]. The agreement is only approximate and the discrepancy between calculated and measured energies are rationalized by the interplay of several approximations. As expected for semi-local DFT models such as PBE functional used here, calculated absolute values of transition energies (e.g., bandgaps) are significant redshifted with respect to experimental values[21]. Moreover, DFT simulations neglect excitonic effects, which are known to be significant in SWCNTs[21]. Finally, a small computational cell is used, leading to some minor spurious defect-defect interactions and the artificial confinement effects. All these simplifications have been chosen to reduce computational cost of simulations of reaction dynamics. There is a qualitative agreement of transition energies between our data and those calculated by more accurate method using similar computational cells (~10 nm SWCNT), time-dependent DFT (TDDFT) that includes excitonic effects, and hybrid long-range corrected functionals[21].

## Data availability

The data supporting the findings of this study are available within the main text or the supplementary information. In addition, Supplementary Data Files 1 and 2 contain atomic coordinates used for band structure calculations and photochemistry modeling, respectively.

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

## Acknowledgements

The authors thanks Ekaterina Dolgopolova and Xiangzhi Li for performing iodoaniline functionalization of (11,0) SWCNT under oxygen-condition and optical experiment, respectively, at LANL. This work was conducted in part at the Center for Integrated Nanotechnologies, an Office of Science User Facility operated for the U.S. Department of Energy (DOE) Office of Science. Los Alamos National Laboratory (LANL), an affirmative action equal opportunity employer, is managed by Triad National Security, LLC for the U.S. Department of Energy's NNSA, under Contract 89233218CNA000001 and supported by DOE, BES, Quantum Information Science Infrastructure Development Project, Deterministic Placement and Integration of Quantum Defect. ST & BG acknowledge partial support from LDRD. DK & YH acknowledge NSF CHE- 1944921. SK & DK thanks the financial support of DOE EPSCoR: Building EPSCoR-State/ National Laboratory Partnerships grant no. DE-SC0021287. M.Z. acknowledges support by NIST internal funding. For computational resources and administrative support, authors thank the Center for Integrated Research Computing (CIRC) at the University of Rochester, the Center for Computationally Assisted Science and Technology (CCAST) at North Dakota State University, and the National Energy Research Scientific Computing Center (NERSC) allocation award ERCAP0021557, supported by the Office of Science of the DOE under contract no. DE-AC02-05CH11231.

## Author contributions

Y.Z. and S.T. conceived the idea of this work. Y.Z. designed the experiments. Nanotube separation, functionalization, and optical characterization were performed by Y.Z. under the supervision of S.K.D. and H.H. Additional purified SWCNT samples were provide by Z.L. and M.Z. Theoretical simulations were performed by Y.H., B.M.W. and B.J.G. under the supervision of D.K., S.K., and S.T. All authors contributed to the analysis and interpretation of results. Y.Z., S.K.D., H.H. and S.T. wrote the manuscript with the assistance of all coauthors.

## Competing interests

The authors declare no competing interests.
