## [Peer Review File · Nature Communications]

Reviewers' Comments:

Reviewer #1 (Remarks to the Author):

I like this paper. It shows that photoexcited aryl groups react in different configurations on nanotube sidewalls than diazonium-based aryl groups with ground-state reactivity. It also shows how dissolved oxygen can impact this reactivity. The reason these results are important are that the reaction configuration affects the single photon emission properties / wavelength of the nanotubes – which is one of their most exciting applications being explored.

The experimental data and simulations are of high quality and complement each other nicely. While the simulations do not exactly predict the experimental results, the agreement is semi quantitative with the right trends observed. The level of theory used has had good predictive power describing localized exciton states in nanotubes previously.

The paper is well written, easy to read, with high quality figures. The nanotube photophysics and chemical functionalization communities will very much appreciate the insights provided paper. It should be accepted after the following revisions:

1) Please conduct control experiments without added aryl molecules but with UV light and with/without oxygen and include the results in the paper. These experiments are important for demonstrating the observed phenomena are occurring due to reaction of photoexcited aryl molecules with the nanotube sidewalls and not some other unexpected reason.

2) Please add to the paper more details of the photochemical reaction conditions. How close was the LED to the solution? What type of vessel was the solution contained in / what area? Any estimate of UV intensity in the nanotube solution? What was the duration of the UV exposure? Please add these details.

3) Please specify the reasoning behind using the 300 nm Au overcoated by PS.

Reviewer #2 (Remarks to the Author):

Zheng et al. report experimental organic color center (OCC) formation for single-walled carbon nanotubes (SWCNTs) by using photoexcited aromatics and conduct first principles simulations to discuss differences in resultant photoluminescence by the reactions in the presence or absence of oxygen in terms of spin multiplicity of photoexcited intermediates. However, the most experimental results, which include photochemical functionalization using aromatic compounds and the generation of E11* (ca. 1100-1200 nm) under air and E11* and E11** (ca. 1200-1300 nm) under argon have been already reported in a published paper from the Weisman group (ACS Nano 2020, 14, 715–723), in which a reaction mechanism based on triplet state aromatics has been also suggested. Accordingly, the simulation part is mainly new achievement of this paper. Therefore, this manuscript would be suitable for other journals specific to computational chemistry.

For further understanding of this system, this reviewer recommends adding discussion comparing to recently reported oxygen doping to SWCNTs by oxygen molecules under UV irradiation from Ao group (J. Phys. Chem. C 2021, 125, 9236–9243), particularly for E11* wavelength difference between argon and oxygen reactions in the authors' result. In addition, the simulation models and results need more consideration. For example, when focusing on the triplet pathway for para binding configuration, dissociation of hydrogen on the 4 position (para) of the ring must occur as the first step in the photoexcited aniline with the triplet state. Adequacy of this reaction process needs to be discussed. The simulation predicts the attachment of the hydrogen and the aryl group from the aniline for defect formation but experimental evidence for this reaction path is needed. Moreover, regarding suggested Pauli exclusion principle, more careful discussion should be conducted to validate this principle application to positional selectivity in the molecular reaction system.

Reviewer #3 (Remarks to the Author):

The authors reported that the photochemically activated reactions between organic aromatic compounds (p-iodoaniline, aniline, and nitrobenzene) and SWCNT provides a controllable route to introduce fluorescent quantum defects in SWCNTs with specific binding configurations. The presence and absence of dissolved oxygen molecules (triplet ground state) in aqueous SWCNTs suspensions controls the binding defects: ortho binding configurations through singlet-state reactions and para configurations through triplet-state reactions, respectively. Particularly, the current paper newly found the para-configurations, which cannot be previously accessible. Controlling the binding configurations allows adjusting SWCNT photoluminescence tunability. The current paper reported significant findings toward practical applications of quantum light sources for quantum information technologies. However, the authors should answer major comments in

fundamental chemistry, before I can recommend the paper for publication in Nature Communications.

Additional Comments:

(1) Details of structural information on the four transition states in Fig. 4 should be given in Supporting Information. In the two transition states (TSs) in ortho binding configurations in the singlet state, a kind of four-centered structures can be found, where a C-H bond of aniline is cleaved, and then the H atom migrates to the nanotube, at the same time, one C-C bond is newly formed between the aryl and nanotube. Although these structures are understandable, these bond lengths should be given.

In the two TSs in para binding configurations in the triplet state, the movie in Supporting information, at first, an H atom suddenly jumps to a para-positioned carbon atom of the nanotube. After that aryl group is attached to a carbon atom of the nanotube. These TSs structures are skeptical for me, in particular the sudden jumping of the H atom. The authors should increase the number of intermediates between the reactant and products as input, or decrease a force tolerance (0.03 eV/Å).

(2) If possible, similar transition states for reactions between aniline and an armchair nanotubes should be obtained, and be compared with those in zigzag nanotubes.

(3) Spin density distribution maps in the four TSs should be given.

(4) In Fig. 3(c), Aryl-NH₂/H and Aryl-NH₂/OH structures were used to calculate the HOMO-LUMO gaps, and then the gaps in the defect structures are compared with those in the pristine structures. It is not clear why the OH group is formed. In addition, information on spin states of Aryl-NH₂/H and Aryl-NH₂/OH structures used for Gaussian calculations should be given.

(5) With respect to Fig. 3, the authors should discuss why the HOMO-LUMO energy shifts ΔE_{HL} link to ΔE_{11} . In other words, the authors should discuss why the HOMO-LUMO gap of defected structures can link to changes of their emission energies, although excited states of defected structures play an important role in their photoluminescence.

(6) One of the main findings in the current paper is to find Aryl-NH₂/H structures in the para-configurations in the triplet spin state from DFT calculations. Can the presence of the triplet-spin states para-configurations be experimentally rationalized ?

Reviewer #4 (Remarks to the Author):

The major claims presented in this manuscript are a) the development of a protocol for photosynthetic control of chemical binding configurations following the photostimulated reaction of certain aromatic molecules with the sidewall of (6,5) carbon nanotubes. Such control is said to be achieved by controlling the molecular oxygen content. This is found to affect the reaction pathway, reportedly allowing to selectively create 'ortho'-, or in the absence of oxygen, the ortho- PLUS previously supposedly inaccessible 'para-' binding-configurations. b) First principles calculations are performed to allow identification of E11* and E11** features with ortho and para binding configurations, respectively. The authors also discuss a possible reaction mechanism based on these calculations.

In principle the topic of this article is timely and is of interest to a broader readership as such research may be of relevance for the development of new single photon emitters. The specific benefit that this research might bring to the field is that para binding configurations provide excited states for single photon emission with lower emission energies than those previously generated by 'ortho' sites.

However, claim a) quoted above has already been published elsewhere. The essentially same type of control over the formation of the E11** state for the same photostimulated reaction appears to have been reported by the lead author and other co-authors in ACS Nano 14 (2020) p715-723 (<https://doi.org/10.1021/acsnano.9b07606>). In the ACS Nano paper the authors report on the light-induced formation of the E11* state when oxygen(air) is present and the additional formation of the E11** state when the reaction takes place in an inert gas atmosphere and oxygen is thus removed.

Moreover, the supposedly 'previously inaccessible para configurations', E11** states, have previously been synthesized with even greater control than in the present study, i.e. selectively. This was achieved using a thermal, not photochemical reaction protocol, see reference 12: Settele et al, Nature Comm. 12 (1), 2119 (<https://doi.org/10.1038/s41467-021-22307-9>). Claim a) thus appears to be correct only for the specific type of photostimulated (!!) reaction protocols, not for other reaction protocols leading to such defects. This distinction may be inferred from the title of the manuscript but not the wording and language used throughout much of the experimental section.

Experimentally new are the ensemble as well as single particle low-temperature PL studies of different functionalized nanotube chiralities which appears to facilitate the identification of the diameter dependence of energies and energy differences between the different defect states. I would argue, however, that despite the obvious experimental challenges, this yields comparatively little new insight. Qualitatively the trends obtained from experiment and DFT theory are similar, but again, the insight gained from the diameter dependence appears to be minor. If the diameter dependence serves a specific purpose, this does not seem to be clearly discussed in the manuscript.

The second claim b), namely the assignment of optical features to structural configurations by theoretical analysis of the reaction mechanism and electronic structure appears to be valid.

Overall, the methodology, interpretation and conclusions appear to be sound. However, owing to the shortcomings discussed above, the authors are asked to thoroughly rephrase key sections of the manuscript to more clearly communicate which aspects of the work are truly new and insightful and which are not.

Minor points of criticism.

- the authors use both, electron-Volt and nanometers as units of measure for PL wavelengths. I would suggest to use the same unit for all energy axis

- the waterfall plots in Fig. 1a obscure subtle but possibly interesting details of the spectra such as peak shape or a dependence or non-dependence of peak emission wavelengths on the degree of functionalization and should thus be replaced by a vertical stack of spectra.

- the details of the reaction conditions in the ACS Nano paper by Zhang et al. and this manuscript appear to be somewhat different. It would be interesting to learn about the thinking behind those changes

The Reviewers comments are given in **black**, the author responses are highlighted in **red**, whereas actual changes in the manuscript are shown in **blue**.

Reviewer #1 (Remarks to the Author):

I like this paper. It shows that photoexcited aryl groups react in different configurations on nanotube sidewalls than diazonium-based aryl groups with ground-state reactivity. It also shows how dissolved oxygen can impact this reactivity. The reason these results are important are that the reaction configuration affects the single photon emission properties / wavelength of the nanotubes – which is one of their most exciting applications being explored.

The experimental data and simulations are of high quality and complement each other nicely. While the simulations do not exactly predict the experimental results, the agreement is semi quantitative with the right trends observed. The level of theory used has had good predictive power describing localized exciton states in nanotubes previously.

The paper is well written, easy to read, with high quality figures. The nanotube photophysics and chemical functionalization communities will very much appreciate the insights provided paper. It should be accepted after the following revisions:

We appreciate high evaluation of our work and thank the Reviewer for summarizing strong points and possible impact of our study.

1) Please conduct control experiments without added aryl molecules but with UV light and with/without oxygen and include the results in the paper. These experiments are important for demonstrating the observed phenomena are occurring due to reaction of photoexcited aryl molecules with the nanotube sidewalls and not some other unexpected reason.

This is fair concern. The control experiment has been done and was reposted in Fig. S10-S12, Supporting information of Ref. 10). Dr. Yu Zheng, the first author of the present MS and previous paper (Ref. 10) has performed these control experiments, e.g. Fig. S12 copied below.

UV irradiation of SDS-coated SWCNTs without added aromatic compounds

Figure S11. Fluorescence spectra for SDS-suspended SWCNTs irradiated with UV light for 0, 1, 2, and 2 hours. Only minor fluorescence intensity changes are evident.

We have introduced an appropriate comment on page 6 of the revised MS:

Control experiments reported previously (Figs. S10-S12 of Supporting Information in Ref.¹⁰) confirm that the covalent SWCNT functionalization indeed requires addition of aromatic compounds followed by UV photoexcitation.

2) Please add to the paper more details of the photochemical reaction conditions. How close was the LED to the solution? What type of vessel was the solution contained in / what area? Any estimate of UV intensity in the nanotube solution? What was the duration of the UV exposure? Please add these details.

We added the following details for the photochemical reaction condition to the Method: Functionalization of suspended SWCNT samples, page 17:

UV LED (Boston Electronics, product code VPC1A1) with emission peaking at 300 (280-320 nm range) was used to irradiate the aromatics/SWCNT mixtures at the power density of ca. 6 mW/cm² at a LED drive current of 0.15A. The aromatics/SWCNT mixtures was hold in a 1cm x 1 cm UV grade quartz cuvette that is placed next to the LED ensuring that entire mixture is illuminated uniformly. The exposure times are varied from 30 s to an hours as indicated in the figures.

3) Please specify the reasoning behind using the 300 nm Au overcoated by PS.

Cover slips with 300 nm Au overcoated by PS is used to enhance the PL collection efficiency and suppress the background. A thick layer of PS (>100 nm) is used so that the Au layer could

not affect the emission properties of SWCNT through plasmonic effects. The following sentence is added to Methods: Preparation of SWCNT substrates page 17.

The polymer coated gold layer is used to enhance the PL collection efficiency without modifying the emission properties of SWCNT through plasmonic effects.

Reviewer #2 (Remarks to the Author):

Zheng et al. report experimental organic color center (OCC) formation for single-walled carbon nanotubes (SWCNTs) by using photoexcited aromatics and conduct first principles simulations to discuss differences in resultant photoluminescence by the reactions in the presence or absence of oxygen in terms of spin multiplicity of photoexcited intermediates. However, the most experimental results, which include photochemical functionalization using aromatic compounds and the generation of E11* (ca. 1100-1200 nm) under air and E11* and E11** (ca. 1200-1300 nm) under argon have been already reported in a published paper from the Weisman group (ACS Nano 2020, 14, 715–723), in which a reaction mechanism based on triplet state aromatics has been also suggested. Accordingly, the simulation part is mainly new achievement of this paper. Therefore, this manuscript would be suitable for other journals specific to computational chemistry.

We humbly disagree with the Reviewer critique. The Reviewer asserts that the bulk of our experimental work simply repeats what was already reported in the Zheng/Bachilo/Weisman ACS Nano 2020, 14, 715–723 paper. This is an inaccurate characterization of the work we present. We agree the results shown in our Fig. 1 in many respects repeats the Weisman work, but this solely serves to introduce the photochemistry we are studying, and demonstrates the spectral behavior our study aims to understand. Consistent with earlier work from our group, the Weisman paper correctly notes the different spectral behaviors arise from formation of different binding configurations, but is unable to definitively assign which configurations are being formed, due to the Weisman work being limited to ensemble level measurements and observation of behavior only for the (6,5) chirality. While their paper also suggests the possible important role that triplet states may play in the contrasting oxygenated vs. de-oxygenated results, it is purely speculative on their part, with no further experimental or theory results given to support their assertion. Thus, while the Weisman paper clearly introduces the new photochemistry of SWCNT defect formation from reactions with photoexcited aromatics, it leaves many open questions that we are able to answer here, based on our much more extensive experimental and theoretical results summarized in the subsequent Figures.

Our Figures 2 and 3 represent a large body of completely new experimental results comprising extensive single-tube low-temperature spectroscopic measurements and ensemble measurements made across multiple pure (n,m) SWCNT samples ranging from near-armchair type (6,5) and (7,6) tubes to near- and pure zigzag (9,1) and (11,0) chiralities. This combination of single-tube and multi-chirality experiments (not appearing in the Weisman work) are essential to demonstrate definitively that different aromatic binding configurations are formed, dependent on oxygenated vs. deoxygenated reaction conditions, allow us to make accurate assignment of the binding configurations, and most significantly show that in the absence of oxygen *para* configurations are formed. It is important to emphasize the latter finding, as it is the first ever

strong experimental evidence for formation of para configurations, and is highly suggestive that to obtain such binding, the reaction must go through a triplet-based pathway.

Our modeling work summarized in Figure 4 gives a strong theory basis to support this expectation. As a cursory note, we strongly disagree with the Referee opinion that ‘simulation’ work is assumed to be always inferior to experiment by definition, and should be reported only in ‘journals specific to computational chemistry’. High quality simulations can be as informative as high quality experiments. Moreover, a seamless combination of experiment and modeling allows achieving deeper insights – the case we believe we have in the present contribution.

To summarize, our work, thus for the first time, demonstrates that the previously inaccessible para configurations can be formed by opening a triplet-based reaction pathway photochemically. These results significantly expand the atomistic understanding of SWCNT defect chemistry and provides a firm experimental and theoretical footing on which to extend, access, and control defect binding and the accompanying spectroscopic response.

To address the Referee critique, we have introduced a brief discussion on the subject to pp. 4 and 6 to reflect connection and differences with previous Weisman report.

This synthetic route - relying on photoexcited chemistry - was first reported by a few of us (see Ref. 10). This work hypothesized that different spectral features arise from the formation of different binding configurations but was unable to definitively assign which configurations were being formed. In this contribution we identify that chemical reactions between (6, 5) SWCNT sidewalls and photoexcited aromatics in the presence of dissolved oxygen generate additional emission features red-shifted by ca. 160 meV (*i.e.*, E_{11}^* transition), whereas the oxygen-free photoreaction gives two distinct pronounced emission bands red-shifted by ca. 140 and 260 meV (*i.e.*, E_{11}^* and E_{11}^{**} transitions, respectively), originating from *ortho* and *para* binding configurations, respectively. The previously unachievable E_{11}^* emission for (11, 0) SWCNTs that arises from the *para* configuration is also obtained through the oxygen-free photochemistry.

These results are consistent with spectral changes for an ensemble of functionalized CoMoCAT SWCNTs prepared using the same synthetic protocol¹⁰.

For further understanding of this system, this reviewer recommends adding discussion comparing to recently reported oxygen doping to SWCNTs by oxygen molecules under UV irradiation from Ao group (J. Phys. Chem. C 2021, 125, 9236–9243), particularly for E_{11}^* wavelength difference between argon and oxygen reactions in the authors’ result.

That work deals only with oxygen doping on exposure of SWCNTs to reactive oxygen species (ROS) generated by exposure of aqueous samples to high-energy UV photons. Control experiments on our system show that the Ao results are not relevant to our work. For example, exposure of our oxygenated SWCNT solutions to 300 nm light, in the absence of the aromatic reactants, shows no generation of E_{11}^* emission features, indicating the 300 nm excitation wavelength used in our photochemistry experiments is not short enough to generate the ROS of the Ao paper.

We have introduced an appropriate comment on page 6 of the revised MS:

Control experiments reported previously (Figs. S10-S12 of Supporting Information in Ref.¹⁰) confirm that the covalent SWCNT functionalization indeed requires addition of aromatic compounds followed by UV photoexcitation. This delineates present SWCNT functionalization from oxygen doping achieved by exposure of nanotubes to reactive oxygen species generated by exposure of aqueous samples to high-energy UV photons [J. Phys. Chem. C 2021, 125, 9236–9243].

In addition, the simulation models and results need more consideration. For example, when focusing on the triplet pathway for para binding configuration, dissociation of hydrogen on the 4 position (para) of the ring must occur as the first step in the photoexcited aniline with the triplet state. Adequacy of this reaction process needs to be discussed. The simulation predicts the attachment of the hydrogen and the aryl group from the aniline for defect formation but experimental evidence for this reaction path is needed. Moreover, regarding suggested Pauli exclusion principle, more careful discussion should be conducted to validate this principle application to positional selectivity in the molecular reaction system.

We thank the Reviewer for the comment and refer to an answer in our response to the Referee 3, who has requested detailed augmentation of our simulation part with additional modeling detailing all configurational intermediates. Unfortunately, experiments cannot directly probe and identify defect configurations on the atomistic level. Otherwise, decade-long studies of chemically doped SWCNTs that only recently culminated in identification of binding configurations, would be greatly simplified. Currently only quantum-chemical models are able to provide direct atomistic insights into these binding configurations. The Pauli exclusion principle is a qualitative explanation why singlet reaction pathway leads to electrons in the near proximity (i.e. *ortho*- configurations) whereas its triplet counterpart takes the electrons apart (i.e., *para*-configurations).

To address the Referee's remark, we have re-phrased the discussion on p. 14

These observations can be rationalized with simple physical arguments: Singlet-state photochemistry involves a pair of electrons of opposite spins that prefer to be in a close proximity on the same orbital to maintain chemical bonding, leading to *ortho* defects anchored to the same C-C bond. In contrast, triplet-state photochemistry involves electrons of co-linear spins that tend to occupy different orbitals and be spatially separated, which results in a formation of bonds on the opposite ends of the ring, being *para* configurations. This qualitatively explains different energetic barriers for singlet and triplet reactions pathways resulting in *ortho* and *para* binding configurations, respectively, conforming to the Pauli exclusion principle.

Reviewer #3 (Remarks to the Author):

The authors reported that the photochemically activated reactions between organic aromatic compounds (*p*-iodoaniline, aniline, and nitrobenzene) and SWCNT provides a controllable route to introduce fluorescent quantum defects in SWCNTs with specific binding configurations.

We thank the Reviewer for nice summary of our work. We would like to highlight these point by adding the following text to p. 15 of the revised manuscript:

It is important to highlight that the binding configuration is a key factor determining the emission wavelength of the fluorescent covalent defect. Our work reports several ways to control the properties of resulting emitters toward on establishing tunability of SWCNT covalent fluorescent defects.

However, the authors should answer major comments in fundamental chemistry, before I can recommend the paper for publication in Nature Communications.

We thank the reviewer for these comments and hope that we have addressed all his questions in the updated version of the manuscript. In particular, several new figures and their descriptions were added to the revision based on inspiring comments of the Reviewer as detailed below.

Additional Comments:

(1) Details of structural information on the four transition states in Fig. 4 should be given in Supporting Information. In the two transition states (TSs) in ortho binding configurations in the singlet state, a kind of four-centered structures can be found, where a C-H bond of aniline is cleaved, and then the H atom migrates to the nanotube, at the same time, one C-C bond is newly formed between the aryl and nanotube. Although these structures are understandable, these bond lengths should be given. In the two TSs in para binding configurations in the triplet state, the movie in Supporting information, at first, an H atom suddenly jumps to a para-positioned carbon atom of the nanotube. After that aryl group is attached to a carbon atom of the nanotube. These TSs structures are skeptical for me, in particular the sudden jumping of the H atom. The authors should increase the number of intermediates between the reactant and products as input, or decrease a force tolerance (0.03 eV/Å).

To address these comments, we added new to this MS Supplemental Figures 11, 12 and 13. These figures provide additional structural information obtained from Nudge Elastic Band method used for transition state (TS) search to get *ortho*- and *para*-configurations along the reaction trajectory taking at smaller steps and showing more intermediates, as requested by the Reviewer. Moreover, we added a new section to Supplemental Information with detailed discussion of these Figures and analyses of structural information.

Main text changes on pp. 12 and 13:

Figure 4b, c summarizes calculated reaction paths and respective transition state (TS) structures that the system passes during CI-NEB calculations. A detailed analyses of reaction paths and TS structures is given in Supplemental Figures 10-13 along with a Supplemental discussion.

These calculated differences in the TS energy barriers are summarized in Fig. 4d for all four defect configurations considered. Interestingly, the TS geometries leading to *para*(++)/(-) defects, which form the new E₁₁* emission peak, have a distinct dynamical feature: the first step in the reaction is that the H atom is transferred to the tube, while the aryl group remains far away (~4 Å) and unbonded (see Fig. 4c and Supplemental Fig. S13). The opposite situation is found

for *ortho*($++$)/($-$) defects that contribute to the previously observed E_{11}^{**} emission peak. Here, both the H and aryl group are close to the tube at their TS, but the aryl bond establishes itself first.

Supplemental information additions:

Analyses of calculated structural dynamics

The processes of formation of a covalent sp^3 -hybridized defect can be described in terms of four key atoms and three interatomic distances between them illustrated in Supplemental Fig. 11. Specifically, a cleavage of C-H bond of aniline is referred in terms of elongation of C_p - H_t distance, while a migration of hydrogen (H_t) from aniline is described in terms of shortening of the C_c - H_t distance with a final formation of the C_c - H_t bond with the nanotube following by a rapid shortening of the C_p - C_c distance resulting in the bond formation between the Aryl-NH₂ and the nanotube.

Supplemental Figures 12 and 13 show the length of C_p - C_c (blue triangles), C_p - H_t (red squares), and C_c - H_t (green circles) interatomic distances along the reaction path resulting in the formation of *ortho* (Supplemental Fig. 12) and *para* defects (Supplemental Fig. 13) calculated for (11,0) SWCNT. The reaction trajectories passing through the transition state (TS) (intermediate #6 in Supplemental Fig. 12) and resulting in *ortho*-configurations of Aryl-NH₂/H attachments to the (11,0) SWCNT are quite similar for both O^- and O^{++} defects. At the earlier stage of the reaction (snapshots from the reactant to the intermediate #5 in Supplemental Fig. 12), a distance from Aryl-NH₂ to H_t (red squares) stays nearly a constant of ~ 1 Å, while the distance between C_p carbon of aryl and C_c carbon of the SWCNT (blue triangles) systematically decreases from 6.5 Å to 3.5 Å. At the TS (intermediate #6) C_p - C_c bond length (blue triangles) is close to its equilibrium value, while the traveling hydrogen H_t is on the halfway between C_p carbon of aryl (~ 1.7 Å, red squares) and C_h carbon of the nanotube (~ 2 Å, green circles). At the last steps of the reaction after TS (the intermediate #7 and the product in Supplemental Fig. 12), the traveling hydrogen completes its departure from aryl and settles at the SWCNT at the *ortho* bonding position.

Analogously to the *ortho* defects, the reaction trajectory for the *para* defects P^- is nearly the same as for P^{++} (Supplemental Fig. 13). However, there is a significant difference in trajectories between *para* and *ortho* defects. For the pathway leading to the *para*-binding, it takes more steps (more intermediates) for H_t to break of aniline and move towards the C_h of the SWCNT, compared to *ortho*-defects. This longer pathway is rationalized by the requirement of a larger distance between C_p and C_h placed at the opposite sides of the carbon ring of the nanotube to provide a *para* attachment of Aryl-NH₂/H to the SWCNT. Therefore, H_t has to move far away from aryl, reaching up to 4 Å of the C_p - H_t distance (intermediate #5 in Supplemental Fig. 13), to end up at the *para* position, than it does for the *ortho* position (~ 2 Å of the C_p - H_t distance for intermediate #6 in Supplemental Fig. 12). Note that due to a larger mass and a moment of inertia, aryl is much less flexible in its movements compared to H_t , and, therefore, is attaching directly to the closest C_c of the nanotube immediately after H_t is closely approaching the C_h carbon of the nanotube.

Thus, there is a qualitative difference between the nature of TSs and mechanisms of reactions between *para* and *ortho* defects. For *para* defects, the hydrogen H_t is approaching the nanotube first followed by aryl attaching to the nanotube at its TS. In contrast, aryl bounds to the

nanotube first followed by H_t attaching to the nanotube at the TS of *ortho* defects. This difference in reaction trajectories requires a significantly higher kinetic energy of H_t to provide *para* defects, compared to the *ortho* defects. This explains rare formation of *para* defects compared to highly probable *ortho* defects at SWCNTs via chemical reactions, as was reported previously⁶. In contrast, photoreactions might provide enough energy to facilitate formations of *para* defects.

Supplemental Figure 11: Relative position of Aryl-NH₂/H group with respect to (11,0) SWNT surface is characterized by positions of four key atoms: A “traveling” hydrogen (H_t), a “parent” carbon (C_p) at the aryl-NH₂ initially bonded to H_t , and two carbons of the nanotube converting from sp^2 - to sp^3 -hybridized bonding due to an adsorption of aryl-NH₂ to the carbon marked as C_c and an adsorption of H_t to the carbon marked as C_h . These key atoms define the following interatomic distances: C_p - C_c showing the distance between two closest carbons of the Aryl-NH₂ and the nanotube (the blue line here and blue triangles in Supplemental Figures 12 and 13), C_p - H_t corresponding to the distance between terminating/traveling hydrogen and the carbon of the aryl-NH₂ (the red line here and red squares in Supplemental Figures 12 and 13), and C_c - H_t defining the distance between the traveling hydrogen and the carbon of the nanotube to which it will be attached at its final state (the green line here and green circles in Supplemental Figures 12 and 13).

Supplemental Figure 12: A sequence of intermediates between the reactant ((11,0) SWCNT and aniline) and the product (SWCNT with attached Aryl-NH₂/H) for O(-) and O(++) defect configurations. The images of intermediates are shown along the tube axis. Three key interatomic distances along the reaction path correspond to the distance between two closest carbons of the Aryl-NH₂ and the nanotube (blue triangles), the distance between terminating/traveling hydrogen and the carbon of the aryl-NH₂ (red squares), and the distance between traveling hydrogen and the carbon of the nanotube to which it will be attached at its final state (green circles).

Supplemental Figure 13: A sequence of intermediates between the reactant ((11,0) SWCNT and aniline) and the product (SWCNT with attached Aryl-NH₂/H) for P(-) and P(++) defect configurations. The images of intermediates are shown along the tube axis. Three key interatomic distances along the reaction path correspond to the distance between two closest carbons of the Aryl-NH₂ and the nanotube (blue triangles), the distance between terminating/traveling hydrogen and the carbon of the aryl-NH₂ (red squares), and the distance between traveling hydrogen and the carbon of the nanotube to which it will be attached at its final state (green circles). The snapshot highlighted by grey corresponds to the transition state (intermediate #5). At the intermediate #5, the bond between the nanotube and aryl is not yet formed and C_p-C_c bond length is far from its equilibrium value (~ 4 Å), while the hydrogen makes a non-trivial path of its migration: Instead of moving along a straight-line towards the SWCNT, it departs far away from both aryl (~4 Å) and the nanotube surface (~5 Å).

(2) If possible, similar transition states for reactions between aniline and an armchair nanotubes should be obtained, and be compared with those in zigzag nanotubes.

We assume that the Reviewer suggests to include calculations of chiral (near-armchair) semiconducting nanotubes rather than true armchair nanotubes, which are always metallic and cannot be efficient optical emitters. Indeed, it is an excellent idea to compare chiral and zigzag SWCNTs. However, it is nearly impossible to include such calculations for chiral tubes due to high computational cost of Nudge Elastic Band method used for transition state (TS) search in this work. It is important to note that because of the symmetry, the size of the primitive unit cell of the semiconducting chiral nanotubes is about 10 times larger than the unit cell of zigzag tubes of a similar diameter. Therefore, the calculation of chiral tube is significantly longer and requires much more computational resources (RAM and CPUs) compared to calculations of zigzag nanotubes. Therefore, we cannot add these calculations to this work. We also would like to note that experimental data presented in our paper do not show any qualitative difference in the optical response between zigzag and chiral SWCNTs if they belong to the same mod-1 or mod-2 class.

To address this comment, we have included a clarifying remark on page 18 of the revised MS Due to numerical expense, such simulations are only currently tractable for (11,0) zigzag nanotube from the family considered above, which has small unit cell.

(3) Spin density distribution maps in the four TSs should be given.

This information is presented in Supplemental Figure 10, which shows the natural spin orbitals (NSOs) generated at the transition state geometry for each configuration illustrated in Fig. 4c at the main text.

Supplementary Figure 10. Natural spin orbitals (NSOs) were generated from at the transition state geometry for each position shown in Fig. 4c using an extended finite SWCNT with correct capping scheme (see supplemental Table 3). The first column of each configuration shows the bare geometry, while the second shows the corresponding NSOs overlaid. The red and blue iso-surfaces correspond to the two NSOs with population equal to 1 and correspond to the diagonalization of the spin density matrix after a ground state calculation with triplet spin multiplicity. In O+, P++, and P-, one can see the spatial separation of the two electrons (i.e., two doublets). In the other defect configurations, the two electrons are spatially mixed and represent more of a triplet state configuration. The natural spin orbitals (NSOs) were calculated using ground state density functional theory information as input (from Gaussian 16⁴) via the Multiwfn package⁵. The transition state geometries from the nudged elastic band calculations (see methods) were taken and converted to finite sized SWCNTs by replicating the unit cell three times, removing the defect atoms in the outer two cells and capping with four methylene groups (see Supplemental Table 3). The SWCNT was then optimized using spin-restricted B3LYP/STO-3G, keeping the center 0.5 nm on either side of the defect fixed. This allowed the boundaries between unit cells to relax. Then a single point calculation using spin-polarized B3LYP/STO-3G was completed in the triplet multiplicity.

(4) In Fig. 3(c), Aryl-NH₂/H and Aryl-NH₂/OH structures were used to calculate the HOMO-LUMO gaps, and then the gaps in the defect structures are compared with those in the pristine structures.

It is not clear why the OH group is formed.

The experiment is performed under aqueous environment. It is reasonable to hypothesize that partially dissociated water is participating in the reaction via ligand exchange process, especially in access of oxygen. Specifically, the reaction pathway of photoexcited aniline can be intercepted by OH⁻ ions from solvent adsorbing to the surface of SWCNT before or after Aryl-NH₂ radical with lost hydrogen approaching to the nanotube. However, under argon conditions, presence of OH⁻ ions in the solution is expected to be strongly reduced, so that mainly the hydrogen detached from aniline approaches to the nanotube surface together with Aryl-NH₂ forming the defect.

A clarifying remark was added to page 3 on the revised MS

Both H- and OH- binding are relevant to experiments typically performed in aqueous environment.

(b) In addition, information on spin states of Aryl-NH₂/H and Aryl-NH₂/OH structures used for Gaussian calculations should be given.

It is important to note that all stable configurations of defects for both Aryl-NH₂/H and Aryl-NH₂/OH cases were processed with the ground state geometry optimization restricted to the singlet spin multiplicity using Gaussian-16 software. Energetically, this is the lowest electronic state of all functionalized nanotubes directly relevant to the experimental spectroscopy.

A clarifying remark was added to page 18 on the revised MS

These simulations provide the ground state optimal geometry and electronic configuration with singlet multiplicity being the lowest electronic state of all functionalized nanotubes directly relevant to the experimental spectroscopy.

(5) With respect to Fig. 3, the authors should discuss why the HOMO-LUMO energy shifts ΔE_{HL} link to ΔE_{11} . In other word, the authors should discuss why the HOMO-LUMO gap of defected structures can link to changes of their emission energies, although excited states of defected structures play an important role in their photoluminescence.

In our previous studies ([Gifford, B. J.; Kilina, S.; Htoon, H.; Doorn, S. K.; Tretiak, S. Exciton Localization and Optical Emission in Aryl-Functionalized Carbon Nanotubes. *J. Phys. Chem. C* **2018**, *122* (3), 1828–1838], [Kilina, S.; Ramirez, J.; Tretiak, S. Brightening of the Lowest Exciton in Carbon Nanotubes via Chemical Functionalization. *Nano Lett.* **2012**, *12* (5), 2306–2312], [He, X.; Gifford, B. J.; Hartmann, N. F.; Ihly, R.; Ma, X.; Kilina, S. V.; Luo, Y.; Shayan, K.; Strauf, S.; Blackburn, J. L.; et al. Low-Temperature Single Carbon Nanotube Spectroscopy of Sp^3 Quantum Defects. *ACS Nano* **2017**, *11* (11), 10785–10796] and [B. Weight, B. Gifford, S. Tretiak, and S. Kilina; *Interplay between Electrostatic Properties of Molecular Adducts and their Positions at Carbon Nanotubes*; *J. Phys. Chem. C* **2021**, *125*, 4785–4793]), we have shown that the HOMO-LUMO energies follow the same qualitative trends as energies of E_{11}^* transitions in covalently functionalized SWCNTs. Similarities in the behavior of the ground and excited state electronic structures are rationalized by a predominant contribution of the HOMO-LUMO pair to the lowest energy excitonic wavefunction of the SWCNT.[3, 9,13] Therefore, the ΔE_{HL} at the ground state mainly follows the trends and characteristics of the low-lying excitons, ΔE_{11}^* [compare Fig. 2 and Fig. 3 in the paper of B. Weight, B. Gifford, S. Tretiak, and S. Kilina; *Interplay between Electrostatic Properties of Molecular Adducts and their Positions at Carbon Nanotubes*; *J. Phys. Chem. C* 2021, *125*, 4785–4793]

A clarifying remark was added to page 19 on the revised MS

The HOMO-LUMO gap mainly follows the trends and characteristics of the low-lying excitons, ΔE_{11}^{*33} . The agreement is only approximate and the discrepancy between calculated and measured energies are rationalized by the interplay of several approximations.

(6) One of the main findings in the current paper is to find Aryl-NH₂/H structures in the para-configurations in the triplet spin state from DFT calculations. Can the presence of the triplet-spin states para-configurations be experimentally rationalized?

We would like to emphasize that the spin multiplicity of reactant (aniline and the SWCNT), product (covalently functionalized SWCNT with aryl-NH₂/H defect), and the system at the transition state (interacting aryl-NH₂ radical, detached H from aniline, and distorted SWCNT) do not need to coincide. Indeed, the energies of the initial reactants (aniline and the SWCNT) and all products (covalently functionalized SWCNT with aryl-NH₂/H ortho- or para-defect) are found to be minimal for a singlet configuration. The energy of a transition state depends on two factors: (1) a pathway leading to either ortho or para configuration and (2) spin multiplicity dictating the TS barrier height.

One main finding of this work is that the pathway leading to *para* configurations has a minimal value of the transition state barrier under the triplet spin multiplicity. This can be interpreted as follows: The intermediate geometry illustrates formation of two spatially separated radicals: $\text{SWCNT} + \text{C}_6\text{H}_5\text{NH}_2 \rightarrow \text{SWCNT-H} + \text{C}_6\text{H}_4\text{NH}_2$. If isolated, each of these radicals has unpaired number of electrons and prefers a doublet, $s=1/2$, $m=2$ spin multiplicity. Upon exposure to each other, the total spin multiplicity of two doublet radicals follows the angular momentum addition rules and can take values of $S=0$ (singlet) or $S=1$ (triplet). We explore and compare both configurations and identify that certain intermediates along the reaction path prefer triplet spin multiplicity as it is more energetically favorable resulting in a *para*-defect as a product.

However, the optically active transitions of SWCNT with either *ortho*- or *para*-defect (the products of photoreactions) are inspected at the singlet configurations, not triplet. Therefore, the experiment is probing only singlet states of the final products of photoreactions – the functionalized SWCNTs with covalent defects either in *para* or *ortho* positions.

A clarifying remark was added to page 7 on the revised MS

We expect that all photochemical reactions ultimately lead to the true ground state with singlet multiplicity for all functionalized SWCNT species, which is later is probed spectroscopically.

Reviewer #4 (Remarks to the Author):

The major claims presented in this manuscript are a) the development of a protocol for photosynthetic control of chemical binding configurations following the photostimulated reaction of certain aromatic molecules with the sidewall of (6,5) carbon nanotubes. Such control is said to be achieved by controlling the molecular oxygen content. This is found to affect the reaction pathway, reportedly allowing to selectively create ‘ortho’-, or in the absence of oxygen, the ortho- PLUS previously supposedly inaccessible ‘para-’ binding-configurations. b) First principles calculations are performed to allow identification of E11* and E11** features with ortho and para binding configurations, respectively. The authors also discuss a possible reaction mechanism based on these calculations.

In principle the topic of this article is timely and is of interest to a broader readership as such research may be of relevance for the development of new single photon emitters. The specific benefit that this research might bring to the field is that para binding configurations provide excited states for single photon emission with lower emission energies than those previously generated by ‘ortho’ sites.

We thank the Referee for a nice summary and an evaluation of possible impact of our study.

However, claim a) quoted above has already been published elsewhere. The essentially same type of control over the formation of the E11** state for the same photostimulated reaction appears to have been reported by the lead author and other co-authors in ACS Nano 14 (2020) p715-723 (<https://doi.org/10.1021/acsnano.9b07606>). In the ACS Nano paper the authors report on the light-induced formation of the E11* state when oxygen(air) is present and the additional formation of the E11** state when the reaction takes place in an inert gas atmosphere and oxygen is thus removed.

A similar concern has been voiced by the Referee 2. Please see our detailed response to Referee 2 on the subject along with the changes to the MS.

Moreover, the supposedly ‘previously inaccessible para configurations’, E11** states, have previously been synthesized with even greater control than in the present study, i.e. selectively. This was achieved using a thermal, not photochemical reaction protocol, see reference 12: Settele et al, Nature Comm. 12 (1), 2119 (<https://doi.org/10.1038/s41467-021-22307-9>). Claim a) thus appears to be correct only for the specific type of photostimulated (!!) reaction protocols, not for other reaction protocols leading to such defects. This distinction may be inferred from the title of the manuscript but not the wording and language used throughout much of the experimental section.

We would like to disagree with the Reviewer critique. The Reviewer claims that the previously inaccessible para configurations were actually previously synthesized (and with more control) in the Settele Nature Comm. Paper (cited as our ref. 12). This is incorrect, as clearly shown in Fig. 6 of the Settele paper, for which those authors discuss their reaction outcomes ONLY in terms of formation of the *ortho* configurations and never consider or discuss the *para* possibility. While the Settele work does impressively demonstrate reaction conditions leading to highly pure E11** spectral features (which we agree presents a remarkable control of the resultant binding configuration), as they state—it is one of the possible *ortho* configurations that is formed. We believe that the confusion on the Reviewer’s part over this assignment may come from the reviewer equating all E11** spectral features to presence of *para* configurations. This assumption is not correct. Depending on the SWCNT chirality (n,m), the E11** spectral feature can be assigned as arising from either the *ortho* or *para* configuration (see our refs. 19, 20, and 22). Proper configurational assignment can only be made by comparing spectral behaviors from a particular reaction across different chiral types (as we demonstrated previously in our ref. 20). The Settele paper does this for (6,5), (7,5) and (10,5) tubes. Based on the similar spectral behaviors they see across this (n,m) range, they correctly assign their E11** to an *ortho* configuration. Importantly, the Settele paper never discusses a triplet mechanism for formation of the defect states: the only result probed in the absence of oxygen was performed on reactions carried out in the dark (no UV excitation) as shown in Figure S18 of Supplemental Information. Additionally, the only low-T single tube results are reported only (6,5) samples, not their other (n,m). In our work (as described in our discussion around Figure 3), by analyzing spectral behaviors across 5 chiralities spanning near-armchair to zigzag structural types, we observe that different spectral patterns to arise, which are consistent with *para* configuration formation.

To clarify this point on interpretation of the configurational origin of specific spectral features, we add the following text to page 4 of the revise MS:

Another promising synthetic avenue proposed in a recent study¹² is a thermal reaction mechanism proceeding under the dark and under UV irradiation. Similar to diazonium chemistry, this enables formation of several *ortho* defect configurations. The use of UV light or dark conditions allows delineation between E11* and E11** emissions, in particular, leading to highly pure E11** spectral features thus demonstrating a remarkable synthetic control of the resultant two *ortho*- binding configurations. While feasibility of *para* defects was suggested by theoretical

simulations²³, practical chemical routes have so far been limited to *ortho* configurations, hindering SWCNT photoluminescence tunability.

Experimentally new are the ensemble as well as single particle low-temperature PL studies of different functionalized nanotube chiralities which appears to facilitate the identification of the diameter dependence of energies and energy differences between the different defect states. I would argue, however, that despite the obvious experimental challenges, this yields comparatively little new insight. Qualitatively the trends obtained from experiment and DFT theory are similar, but again, the insight gained from the diameter dependence appears to be minor. If the diameter dependence serves a specific purpose, this does not seem to be clearly discussed in the manuscript.

We should note that the diameter dependence of the defect-state emission energies is discussed at length in a number of other works. We include the d-dependence of the photochemically-derived emission energies in Figs. 3b and 3c primarily as useful information for the reader, but it is not the focus of our analysis. As noted in several places above, and in our response to Reviewer 2, however, our studies over multiple chiralities and at the single tube level are not aimed at gaining new insight into diameter-dependent behavior. Far more importantly, the multiple (n,m) and single-tube results allow us to make accurate configurational assignment of our spectral features, which then strongly support our claim of first time generation of *para* configurations, and motivates our theory probing of the triplet role in the reaction chemistry.

We have further emphasized this point on p. 9 in the revised MS

By considering a diverse family of SWCNTs, we aim to associate spectral features appearing due to photochemical reactions with that formed through conventional diazonium chemistry (which underpin *ortho* configurations), as well as to identify additional unique emissive features stemming from *para* defects.

The second claim b), namely the assignment of optical features to structural configurations by theoretical analysis of the reaction mechanism and electronic structure appears to be valid.

Overall, the methodology, interpretation and conclusions appear to be sound. However, owing to the shortcomings discussed above, the authors are asked to thoroughly rephrase key sections of the manuscript to more clearly communicate which aspects of the work are truly new and insightful and which are not.

Hopefully our detailed response the Referees comments eliminates possible misunderstanding and our revisions to the article inspired by this discussion further clarifies our message to the Reader.

Minor points of criticism.

- the authors use both, electron-Volt and nanometers as units of measure for PL wavelengths. I would suggest to use the same unit for all energy axis

This is valid critique. Thanks a lot. We have unified all plots to eV units in both MS and its SI.

- the waterfall plots in Fig. 1a obscure subtle but possibly interesting details of the spectra such as peak shape or a dependence or non-dependence of peak emission wavelengths on the degree of functionalization and should thus be replaced by a vertical stack of spectra.

Figure 1 was re-plotted to address this critique.

- the details of the reaction conditions in the ACS Nano paper by Zhang et al. and this manuscript appear to be somewhat different. It would be interesting to learn about the thinking behind those changes

It is not clear what difference in reaction conditions the reviewer refers to. Reactions in the current work are run in as nearly identical conditions as the work of Zhang/Weisman as we could duplicate. The LED light source and exposure conditions (see also above) are essentially the same. Additionally, the same reactant concentrations and SWCNT reaction volume are used (0.5 mM - 3 mM of aromatics in the Zhang work, 0.5 mM - 1 mM here). One point of difference is that the work of Zhang/Weisman did not incorporate use of purified, single chirality SWCNT samples, while we do in our case. As noted above, such samples are essential for enabling the clear spectral interpretation and assignment of PL features to binding configurations that allow us to make the described claims on triplet reaction chemistry.

REVIEWER COMMENTS

Reviewer #1 (Remarks to the Author):

I originally missed the Weisman ACS Nano 2020, 14, 1, 715–723 paper. It was buried in the wrong places. I went back and reread the previous version of the manuscript, and the manuscript definitely reads as if it is claiming to have invented a “new synthetic protocol” as if the Weisman paper had never been published.

For example, from the original manuscript,

“Here, we demonstrate a new synthetic protocol to control binding configurations and create the previously inaccessible para geometry of quantum defects in SWCNTs through a precise photochemical functionalization.” &

“Chemical reactions between (6, 5) SWCNT sidewalls and photoexcited aromatics in the presence of dissolved oxygen have been found to generate an additional emission features red-shifted by ca. 160 meV (i.e., E11* transition), whereas the oxygen-free photoreaction gives two distinct pronounced emission bands red-shifted by ca. 140 and 260 meV (i.e., E11* and E11** transitions, respectively)”

should have both cited the Weisman paper! This was big omission, and I can see why the other reviewers were up in arms!

--

With this said, I agree with the authors comments in the rebuttal letter that there are significant new results and insights presented. For example, the response to Reviewer 2, Comment 1 is strong. Ignoring the omission in the previous version of the manuscript and focusing on the current version, I still think it warrants acceptance and publication into Nature Comm. The assignment of the binding configuration via theory adds so much above and beyond Weisman ACS Nano 2020.

The revisions have addressed my previous questions. The one major change needed, however, is that the paper still needs to better rephrase itself and better cite Weisman ACS Nano 2020.

For example, the abstract overstates itself: “Here, we develop a new photosynthetic protocol capable of controlling chemical binding configurations of quantum defects, which are often referred to as organic color centers, through the spin multiplicity of photoexcited intermediates.”

The new paper did not develop the synthetic protocols. Rather, the paper has understood the results of synthetic protocols already reported.

Also, in line 79, “we demonstrate a synthetic protocol to” still sounds like the new paper is the first to report such a protocol. Again, the paper has understood the results of synthetic protocols already reported.

Reviewer #2 (Remarks to the Author):

The authors’ reply and revision could show a difference from previous work in ACS Nano. For a publication of this paper, the following revisions are needed.

Additional simulation studies are supportive to understand more details about the reaction pathways in this research. Still, one unclear point is the reason why hydrogen detachment from the aniline compound needs to occur before the reaction with SWCNTs in terms of a chemical reaction path of radical formation from the photo-excited aniline compounds with triplet states. The yield (efficiency) of this hydrogen detachment reaction may relate to the experimental results of both of E11* and E11** emission observation for all functionalized SWCNTs prepared under argon (oxygen-free) conditions. Because the triplet state formation of the aniline compounds and their hydrogen detachment could be a decisive factor for the resultant binding configurations. Therefore, further discussion regarding this point needs to be added in the manuscript.

As the authors pointed out, multi-chirality experiments are a new part compared to the previous work. Thus, the emission spectra of (9,1) and (11,0) SWCNTs treated using iodoaniline under oxygen-conditions need to be added in Supplementary Information. Confirmation of selective E11** generation for these near zigzag- and zigzag tubes by the authors’ method would support the suggested radical reaction pathway that are similarly taken in previous aryldiazonium reaction systems (Ref. 22).

Reviewer #3 (Remarks to the Author):

The authors provided additional information on structural changes during the reaction between (11,0) tube and Aryl-NH₂ (Supplemental Figure 12 and 13) as the reply for my comment (1). Despite the newly added information, I again think that the TSs in triplet spin state are skeptical, because of the sudden jumping of the H atom of the aryl group to nanotube, which can be seen in Supplemental Figure 13 between intermediates #4 to #6. I found that further increasing the number of images of intermediates in NEB calculations is indispensable to obtain TSs in the triplet state to smoothly connect between the reactant and products. This would influence discussion on relative stability between singlet and triplet spin states in the TS. Furthermore, the authors cannot experimentally observe intermediates in the triplet state during the reaction to form the para-configurations, although the final product (Aryl-NH₂/H structure) was observed to be in the singlet state. This information corresponds to the reply for my comment (6). Thus, I found that results obtained experimentally and theoretically cannot support the conclusion in the paper (the reaction forming Aryl-NH₂/H structures in the para-configurations proceeds in the triplet spin state). Accordingly, I cannot recommend the paper for publication in Nature Communication.

Reviewer #4 (Remarks to the Author):

The authors have responded at length to the criticism of the reviewers. Their answers, in addition to the modifications made in the manuscript do provide a new level of insight and detail that is helpful for understanding some of the more salient points.

This reviewer's primary critique, that the first major claim of this work has essentially been published elsewhere appears to have been refuted.

I have no further points of criticism.

The Reviewers comments are given in **black**, the author responses are highlighted in **red**, whereas actual changes in the manuscript are shown in **blue**.

Reviewer #1 (Remarks to the Author):

I originally missed the Weisman ACS Nano 2020, 14, 1, 715–723 paper. It was buried in the wrong places. I went back and reread the previous version of the manuscript, and the manuscript definitely reads as if it is claiming to have invented a “new synthetic protocol” as if the Weisman paper had never been published.

For example, from the original manuscript,

“Here, we demonstrate a new synthetic protocol to control binding configurations and create the previously inaccessible para geometry of quantum defects in SWCNTs through a precise photochemical functionalization.” &

“Chemical reactions between (6, 5) SWCNT sidewalls and photoexcited aromatics in the presence of dissolved oxygen have been found to generate an additional emission features red-shifted by ca. 160 meV (i.e., E11 transition), whereas the oxygen-free photoreaction gives two distinct pronounced emission bands red-shifted by ca. 140 and 260 meV (i.e., E11 and E11 transitions, respectively)” should have both cited the Weisman paper! This was big omission, and I can see why the other reviewers were up in arms!

With this said, I agree with the authors comments in the rebuttal letter that there are significant new results and insights presented. For example, the response to Reviewer 2, Comment 1 is strong. Ignoring the omission in the previous version of the manuscript and focusing on the current version, I still think it warrants acceptance and publication into Nature Comm. The assignment of the binding configuration via theory adds so much above and beyond Weisman ACS Nano 2020.

The revisions have addressed my previous questions.

We appreciate that the Reviewer recognized the benefits of our previous revision and thank the Reviewer for emphasizing novelty our study.

1) The one major change needed, however, is that the paper still needs to better rephrase itself and better cite Weisman ACS Nano 2020.

For example, the abstract overstates itself “Here, we develop a new photosynthetic protocol capable of controlling chemical binding configurations of quantum defects, which are often referred to as organic color centers, through the spin multiplicity of photoexcited intermediates.”

We fully agree with the Referee. These changes were made in the current revision.

Here, we explore recently reported photosynthetic protocol and find that it can control chemical binding configurations of quantum defects, which are often referred to as organic color centers, through the spin multiplicity of photoexcited intermediates.

2) The new paper did not develop the synthetic protocols. Rather, the paper has understood the results of synthetic protocols already reported.

Also, in line 79, “we demonstrate a synthetic protocol to” still sounds like the new paper is the first to report such a protocol. Again, the paper has understood the results of synthetic protocols already reported.

Again, we fully agree with the Referee. These changes were made in the current revision.

Here, we demonstrate that recently developed synthetic protocol is capable of controlling binding configurations by accessing the previously inaccessible *para* geometry of quantum defects in SWCNTs through a precise photochemical functionalization.

Reviewer #2 (Remarks to the Author):

The authors’ reply and revision could show a difference from previous work in ACS Nano. For a publication of this paper, the following revisions are needed.

Additional simulation studies are supportive to understand more details about the reaction pathways in this research. Still, one unclear point is the reason why hydrogen detachment from the aniline compound needs to occur before the reaction with SWCNTs in terms of a chemical reaction path of radical formation from the photo-excited aniline compounds with triplet states. The yield (efficiency) of this hydrogen detachment reaction may relate to the experimental results of both of E11 and E11 emission observation for all functionalized SWCNTs prepared under argon (oxygen-free) conditions. Because the triplet state formation of the aniline compounds and their hydrogen detachment could be a decisive factor for the resultant binding configurations. Therefore, further discussion regarding this point needs to be added in the manuscript.

We thank the Referee for bringing forward this point. Hydrogen detachment is a direct outcome of our computational modeling. We note that our results are fully consistent with the mechanism of many well-documented small-molecule photoaddition reactions. See for instance, reaction (10.1) in p362, “Modern Molecular Photochemistry” by Nicholas J. Turro (University Science Books, 1991):

In the revision, we have added the above discussion to SI and a remark along with new citation of Turro’s book.

Page 13:

Observed hydrogen detachment is a direct outcome of our computational modeling. These results are fully consistent with the mechanism of many well-documented small molecule photoaddition reactions (for example, see Supplemental Scheme 1).

Supplemental information additions:
Page 22

Supplemental Scheme 1: The hydrogen detachment mechanism in small molecule photoaddition reactions: shown is reaction (10.1) in p362 of Ref. 7.

As the authors pointed out, multi-chirality experiments are a new part compared to the previous work. Thus, the emission spectra of (9,1) and (11,0) SWCNTs treated using iodoaniline under oxygen-conditions need to be added in Supplementary Information. Confirmation of selective E₁₁* generation for these near zigzag- and zigzag tubes by the authors' method would support the suggested radical reaction pathway that are similarly taken in previous aryldiazonium reaction systems (Ref. 22).

We have spent substantial amount of time (few months) to functionalize (9,1) and (11,0) at both LANL and NIST labs, and observed a strong chiral angle dependent reaction outcome:

1. Reaction of near-armchair (6,5) with iodoaniline in air readily produces strong E₁₁* emission at 1124 nm (ortho configuration);
2. Reaction of (8,3) (with a chiral angle between armchair and zigzag) under the same condition gives much weaker E₁₁* emission at 1130 nm;
3. Reaction of zigzag (11,0) as well as near-zigzag (9,1) under the same condition yields non measurable E₁₁*.

These results are not new. In fact, similar chiral angle dependence was first shown in ref 18 (ACS Nano 2020, 14, 715–723) Figure 2e.

These observations do not contradict the spin-state control mechanism we study in this contribution. However, they do suggest presence of an additional mechanism by which the reaction

outcome becomes very much dependent on chiral angle. This mechanism and the chirality trend cannot be reliably elucidated at the moment given low emission yield and a minimal amount of sample materials available for experiment. We do appreciate the Referee's comment and intend to explore this emission trend with chirality in the future.

In the revised SI, we have added the above figure and relevant discussion along with an appropriate reference to these results from the Discussion section of the MS.

P. 15

Finally, we point out a strong chiral angle dependent photochemical reaction outcome (see Supplemental Fig. 17 and Fig. 2e in Ref. 18), which suggests presence of an additional mechanism. These trends will be explored in depth in the future studies.

Supplemental information additions:

Page 23

Supplemental Figure 17: A strong chiral angle dependent reaction outcome with iodoaniline in air: a strong E_{11}^* emission at 1124 nm (ortho configuration) in near-armchair (6,5) SWCNT becomes weaker in the intermediate case in (8,3) tube at 1130 nm and finally is non-measurable in a zigzag (11,0) tube.

Reviewer #3 (Remarks to the Author):

The authors provided additional information on structural changes during the reaction between (11,0) tube and Aryl-NH₂ (Supplemental Figure 12 and 13) as the reply for my comment (1). Despite the newly added information, I again think that the TSs in triplet spin state are skeptical, because of the sudden jumping of the H atom of the aryl group to nanotube, which can be seen in Supplemental Figure 13 between intermediates #4 to #6. I found that further increasing the number of images of intermediates in NEB calculations is indispensable to obtain TSs in the triplet state to smoothly connect between the reactant and products. This would influence discussion on relative stability between singlet and triplet spin states in the TS.

To confirm the transition states in *para*-configurations, we carry out two additional calculations.

First, we perform spin-restricted DFT based *ab initio* molecular dynamics (AIMD) calculations using intermediate structures obtained from CI-NEB in Supplemental Figure 13. Such calculations are performed at 100 K with a time step of 1 fs. One finds the smooth connection between initial

reactant (CNT + aniline) and the final product (*para*(-) or *para*(++) configuration) along the AIMD trajectories. We then carry out single point energy calculations by spin-polarized DFT for all snapshots along the AIMD trajectories, from which energy profiles for snapshots through singlet and triplet pathways are obtained (Supplemental Figure 14a). In Figure 14b, we track three key interatomic distances along the trajectories. Note the energy profiles and evolution of interatomic distances are similar for *para*(-) and *para*(++) configurations, due to the symmetry of these conformations. We find an increase in total energy of snapshots when the hydrogen starts to get ejected from the aniline molecule and migrates to the tube. Figure 14c shows snapshots along the trajectory for *para*(-) corresponding to the region between images #4 and #6 in Supplemental Figure 13. Starting from 74 fs (the third snapshot from the left in the second row of Figure 14c), one finds the total energy of a triplet state is becoming lower than that of a singlet state. This trend remains until 132 fs when the hydrogen migration completes, and the aryl group gets closer to the tube. The TS state predicted by CI-NEB (image #5 in Supplemental Figure 13 corresponds to the third snapshot from the left in the third row of Figure 13c).

Second, we perform CI-NEB calculations with 15 images of intermediates (Supplemental Figures 15 and 16). Since the reaction pathways for *para*(-) and *para*(++) are similar, we only focus on the former to save numerical resources. Our numerical results confirm that the reactant (images # 1-8, 14-15) and the product are in a singlet state, whereas images #9-13 are in a triplet state. The TS state is at image #10 with an energy of 4.944 eV, which is close to the TS energy of 4.943 eV obtained from the CI-NEB calculation with 7 images. The TS structures obtained by 15 images and 7 images are similar, except that the ejected hydrogen is becoming about 0.7 Å closer to the binding site on the tube surface for the former.

We rationalize the obtained results by the difference in the electron-vibrational couplings between triplet and single electronic states. The triplet electronic state has a strong coupling with the vibrational mode corresponding to the stretching of the H-C bond of aryl, compared to the singlet state. This results in a stronger distortion of the C-H bond of aryl in its triplet state leading to the transitional state involving detached H from the aryl group and bonding to the nanotube. Then the aryl radical is attaching to the nanotube in the *para* position without any significant changes in its location with respect to the CNT with bonded H, due to its large moment of inertia.

Page 12:

A detailed analyses of reaction paths and TS structures is given in Supplemental Figs. 10-16 along with a Supplemental Discussion.

Supplemental information additions:

Page 15

To further explore the transition states in *para*-configurations, we carry out two additional calculations. First, we perform spin-restricted DFT based *ab initio* molecular dynamics (AIMD) calculations using intermediate structures obtained from CI-NEB in Supplemental Fig. 13. Such calculations are performed at 100 K with a time step of 1 fs. One finds the smooth connection between initial reactant (CNT + aniline) and the final product (*para*(-) or *para*(++) configuration)

along the AIMD trajectories. We then carry out single point energy calculations by spin-polarized DFT for all snapshots along the AIMD trajectories, from which energy profiles for snapshots through singlet and triplet pathways are obtained (Supplemental Fig. 14a). In Fig. 14b, we track three key interatomic distances along the trajectories. Note the energy profiles and evolution of interatomic distances are similar for *para*(-) and *para*(++) configurations, due to the symmetry of these conformations. We find an increase in total energy of snapshots when the hydrogen starts to get ejected from the aniline molecule and migrates to the tube. Figure 14c shows snapshots along the trajectory for *para*(-) corresponding to the region between images #4 and #6 in Supplemental Fig. 13. Starting from 74 fs (the third snapshot from the left in the second row of Fig. 14c), one finds the total energy of a triplet state is becoming lower than that of a singlet state. This trend remains until 132 fs when the hydrogen migration completes, and the aryl group gets closer to the tube. The TS state predicted by CI-NEB (image #5 in Supplemental Fig. 13 corresponds to the third snapshot from the left in the third row of Fig. 13c).

Second, we perform CI-NEB calculations with 15 images of intermediates (Supplemental Figs. 15 and 16). Since the reaction pathways for *para*(-) and *para*(++) are similar, we only focus on the former to save numerical resources. Our numerical results confirm that the reactant (images # 1-8, 14-15) and the product are in a singlet state, whereas images #9-13 are in a triplet state. The TS state is at image #10 with an energy of 4.944 eV, which is close to the TS energy of 4.943 eV obtained from the CI-NEB calculation with 7 images. The TS structures obtained by 15 images and 7 images are similar, except that the ejected hydrogen is becoming about 0.7 Å closer to the binding site on the tube surface for the former.

We rationalize the obtained results by the difference in the electron-vibrational couplings between triplet and single electronic states. The triplet electronic state has a strong coupling with the vibrational mode corresponding to the stretching of the H-C bond of aryl, compared to the singlet state. This results in a stronger distortion of the C-H bond of aryl in its triplet state leading to the transitional state involving detached H from the aryl group and bonding to the nanotube. Then the aryl radical is attaching to the nanotube in the *para* position without any significant changes in its location with respect to the CNT with bonded H, due to its large moment of inertia.

Supplemental Figure 14: (a) Calculated energy profiles along the DFT-based AIMD trajectories, which convert the reactant ((11,0) SWCNT and aniline) into the product (SWCNT with attached Aryl-NH₂/H) for *para*(-) and *para*(++) defect configurations. (b) Three key interatomic distances along the trajectories correspond to the distance between two closest carbons of the Aryl-NH₂ and the nanotube, the distance between terminating/traveling hydrogen and the carbon of the aryl-NH₂, and the distance between traveling hydrogen and the carbon of the nanotube to which it will be attached at its final state. (c) The snapshots in the region defined by dashed grey lines in panels (a) and (b) for the AIMD trajectory leading to *para*(-) configuration shown along the tube axis. The AIMD calculations are based on images of intermediates from CI-NEB calculations. The AIMD calculations are performed at 100 K with a time step of 1 fs.

Supplemental Figure 15: (a) Calculated minimum energy path leading to *para*(-) binding configuration by CI-NEB with 15 images of intermediates. Structures with open circles (the reactant, images #1-8, 14-15, and the product) are in a singlet state, whereas those with closed circles (images #9-13) are in a triplet state. (b) comparison of TS states from CI-NEB with 15 images of intermediates and with 7 images of intermediates (Supplemental Figure 13).

Supplemental Figure 16: A sequence of intermediates between the reactant ((11,0) SWCNT and aniline) and the product (SWCNT with attached Aryl-NH₂/H) for *para*(-) defect configuration. The images of intermediates are shown along the tube axis. Three key interatomic distances along the reaction path correspond to the distance between two closest carbons of the Aryl-NH₂ and the nanotube (blue triangles), the distance between terminating/traveling hydrogen and the carbon of the aryl-NH₂ (red squares), and the distance between traveling hydrogen and the carbon of the nanotube to which it will be attached at its final state (green circles). The snapshot highlighted by grey corresponds to the transition state (intermediate #10).

Reviewer #4 (Remarks to the Author):

The authors have responded at length to the criticism of the reviewers. Their answers, in addition to the modifications made in the manuscript do provide a new level of insight and detail that is helpful for understanding some of the more salient points.

This reviewer's primary critique, that the first major claim of this work has essentially been published elsewhere appears to have been refuted.

I have no further points of criticism.

We appreciate that the Referee has recognized the value of our previous revision.

REVIEWERS' COMMENTS

Reviewer #2 (Remarks to the Author):

The authors addressed most of my concerns. However, there is one comment regarding the added Supplemental Scheme 1. The described reaction scheme would mention that photoexcited ketone group (n, π^* state) in the RC=OR compound proceeds with the hydrogen abstraction from the X-H compound to produce the radical pair. In contrast, in the authors' system, the aniline compound is photoexcited and the hydrogen attachment occurs on the nanotube side. Therefore, the described reaction path in Supplemental Scheme 1 is different from the proposed reaction system in this study. Accordingly, other references and a consistent scheme need to be cited for this discussion.

Reviewer #3 (Remarks to the Author):

The authors performed additional DFT calculations (DFT-based ab initio MD calculations & NI-NEB calculations with 15 images), and properly answered the questions that I asked. Now, I am satisfied with the corrections. Accordingly, I can recommend the revised paper for publication in Nature Communication.

Reviewer #2 (Remarks to the Author):

The authors addressed most of my concerns. However, there is one comment regarding the added Supplemental Scheme 1. The described reaction scheme would mention that photoexcited ketone group (n,π^* state) in the RC=OR compound proceeds with the hydrogen abstraction from the X-H compound to produce the radical pair. In contrast, in the authors' system, the aniline compound is photoexcited and the hydrogen attachment occurs on the nanotube side. Therefore, the described reaction path in Supplemental Scheme 1 is different from the proposed reaction system in this study. According, other references and a consistent scheme need to be cited for this discussion.

Response: The referee is correct that the situation described in Supplementary Scheme 1 is not exactly the same as our reaction. But the idea conveyed by this Scheme 1, i.e., a double bond turning into a radical with the extraction of a hydrogen, is still a support to our mechanism invoking H-extraction. To address the Referee comment we deleted Supplementary Scheme 1 from SI and included a general citation of the photochemistry book in the main text on page 13: "These results are fully consistent with the mechanism of many well-documented small molecule photoaddition reactions [50]."